


# An updated EAWS matrix to determine the avalanche danger level: derivation, usage, and consistency

Karsten Müller[1,*], Frank Techel[2,*], and Christoph Mitterer[3]

[1]Norwegian Water Resources and Energy Directorate, Oslo, Norway
[2]WSL Institute for Snow and Avalanche Research SLF, Davos, Switzerland
[3]Avalanche Warnings Service Tirol, Innsbruck, Austria
[*]Joint first authors. These authors contributed equally to this work.

**Correspondence:** Karsten Müller (kmu@nve.no)

**Abstract.** Avalanche forecasting plays a crucial role in mitigating risks associated with snow avalanches in mountainous regions. Standards for regional avalanche forecasting were initially developed at national levels. Therefore, the introduction of the European Avalanche Danger Scale (EADS) in 1993, still in use today, represented a milestone in harmonizing the assessment and communication of avalanche danger. However, standards, concepts and definitions have evolved since then. Here, we

reflect on the current standards and definitions used in regional avalanche forecasting, with a focus on the updated European Avalanche Warning Services (EAWS) Matrix, a look-up table intended to ensure consistency among avalanche forecasters when assigning a danger level. The EAWS Matrix links the factors determining avalanche danger - snowpack stability, the frequency of snowpack stability, and avalanche size - to avalanche danger levels. Here, we describe the methodology to obtain a consensus-based EAWS Matrix. Finally, by analyzing the operational use of the EAWS Matrix following its introduction, we

gain insights into its implementation across European avalanche warning services and obtain an understanding on challenges and short-comings related to its operational use. As a reliable estimation of the factors determining avalanche danger is a prerequisite for consistency in assigning the danger levels using the Matrix, we also explored the consistency of estimating the factors by comparing forecasts prepared by individual forecasters. Noting considerable variations in the assignment of factor classes, we provide recommendations for practice and ways forward, such as refining the definitions of the classes describing

the factors, implementing training sessions, and exploring different matrix layouts. Additionally, the discrepancies between the EADS and current standards and definitions underscore the need for an updated avalanche danger scale. In conclusion, the updated EAWS Matrix represents a next step towards harmonizing avalanche forecasting practices in Europe even though the analysis revealed areas for improvement. Clearly, further efforts are required to develop and implement regional avalanche forecasting standards to reach the goal of avalanche forecasts being a reliable, credible, and timely source of information of

expected avalanche conditions, regardless of the forecaster or warning service behind the product.



# 1 Introduction

Snow avalanches represent a natural hazard in snow-covered, mountainous regions. Avalanches may lead to injury or loss of life, and can cause damage or destroy property and infrastructure. For instance, in Europe in the 50 years between 1974 and 2023, more than 5900 (annual mean: 118) people have died in avalanches (EAWS, 2023a). To reduce adverse effects resulting from avalanches, avalanche warning services disseminate regional avalanche forecasts to inform and warn the general public as well as responsible decision-makers in local authorities on prevailing and expected avalanche danger valid for a specified region and period. The preparation of these regional avalanche forecasts involves two steps:

1. Assessment of current and future avalanche danger.

2. Communication of future avalanche danger.

The assessment of current and future avalanche conditions involves the analysis of a wide variety of heterogeneous data, including field observations, measurements, and weather forecasts. Although the interpretation of snow and weather parameters follows a deterministic cause-and-effect approach, actual forecasting decisions are reached using inductive logic (LaChapelle, 1980). Thus, the quality of avalanche forecasts is influenced by a combination of factors, including the forecaster's experience and reliability (Stewart and Lusk, 1994; McClung, 2002), as well as the dynamic nature of the snowpack, which varies spatially and temporally (Schweizer et al., 2008). Due to the inherent uncertainty in predicting the exact timing and location of avalanche events, the assessment of avalanche danger maintains a qualitative character. Unlike disciplines such as weather forecasting, where predictions often involve precise numerical values such as temperature or precipitation, avalanche forecasts primarily rely on categorical information. For communication purposes, both between forecasters and between forecasters and forecast users, the complex and multifaceted nature of avalanche conditions is simplified into symbolic representations, encompassing danger levels, classes, terms, and text (Hutter et al., 2021). In regional avalanche forecasting, for instance, the severity of expected avalanche conditions is summarized using the concept of danger levels. Despite advances in model-driven predictions of avalanche danger (e.g., Giraud, 1992; Pérez-Guillén et al., 2022), assessing avalanche danger (levels) has so far remained primarily a subjective decision-making process. While complete consensus between individual forecasters is unattainable, random variations inherent to human judgment should be reduced to a minimum. A consistent representation between a forecasters best judgment and the forecasts they produce is as important as consistency between forecasters, as these directly impact the quality of avalanche forecasts (Murphy, 1993; Stewart, 2001), which in turn increases the value of the forecast to decision-makers using avalanche forecasts as input (Murphy, 1993). Moreover, reliable forecasts contribute to reducing damage and loss caused by avalanches. Thus, striving for increased consistency in avalanche forecasting is imperative for enhancing safety and mitigating risks associated with avalanche hazards.

The information provided in avalanche forecasts is structured following an information pyramid, with the most relevant information, a danger level ($D$), at the top (EAWS, 2023d). The importance of $D$ for decision-making in avalanche terrain has been shown in numerous studies, including, for instance, during the trip planning stage (e.g., Morgan et al., 2023), impacting





the decision whether to ski a slope or not (e.g., Furman et al., 2010), or the correlation between the forecast danger level and avalanche risk during back-country skiing (e.g., Techel et al., 2015; Winkler et al., 2021).

Given the importance of the avalanche danger levels to support decision making for users of avalanche forecasts, ensuring a consistent assignment of these levels is paramount. However, several studies have shown considerable variation in the use of danger levels. These variations are greater between forecasters from different or neighbouring warning services (Lazar et al., 2016; Techel et al., 2018) than within a single warning service (Techel et al., 2018; Lucas et al., 2023). Additionally, inconsistencies persist when describing dry- and wet-snow avalanche conditions in terms of the likelihood and size of natural avalanches (Clark, 2019; Hutter et al., 2021).

With the aim to increase consistency between forecasters and warning services when assessing regional avalanche danger, a working group of the European Avalanche Warning Services (EAWS) revised the definitions of the factors determining regional avalanche danger and developed a common workflow for assessing the avalanche danger level (EAWS, 2022b; Müller et al., 2023). Moreover, the look-up table assisting forecasters in assigning a danger level, referred to as EAWS Matrix, was revised, to be in line with the definitions and terminology used to describe avalanche danger by the working group. Definitions, workflow and matrix were accepted by the EAWS General Assembly in 2022 (EAWS, 2022b).

It is our objective to reflect on current standards and definitions used in regional avalanche forecasting. We do so in three steps, with a focus on the updated EAWS Matrix. First, we summarize the development and definitions of the major standards used in avalanche forecasting in Europe and North America, with an emphasis on their benefits and shortcomings (Sect. 2). This is followed by describing the methodology and outcome of the revised EAWS Matrix (Sect. 3). Finally, analysing the use of the revised EAWS Matrix after the first winter following its introduction (Sect. 4.1), we gain insights on potential challenges and short-comings of using the EAWS Matrix, enabling a further refinement of the standards describing the process of regional avalanche forecasting and fostering the discussion towards an updated European avalanche danger scale.

## 2 Background

### 2.1 The European Avalanche Danger Scale

Avalanche bulletins have been published since the winter 1945/1946 in Switzerland. Although neither standardized nor defined nor used in a consistent manner, avalanche danger was already described in winter 1951/1952 in Switzerland as being *low*, *moderate*, *considerable*, *high* and *very high*, sometimes in connection with modifiers like *general* and *local* (e.g., SLF, 1953, p.68 ff). A first description of the danger levels used in Switzerland was published in 1985 (Föhn, 1985), allowing consistent use by forecasters and transparent communication to users. Similarly, in France, eight «typical» avalanche situations were used to assess and communicate avalanche conditions (Giraud et al., 1987). These were later on also used in Italy. Despite the formation of a European Avalanche Warning Services (EAWS) working group in 1983, which aimed at fostering cooperation across national borders, the Alpine countries France, Italy, Switzerland, Germany and Austria continued to use their own danger scales with a varying number of six to eight danger levels (Mitterer and Mitterer, 2018). In 1993, the EAWS introduced the five-level European Avalanche Danger Scale (EADS, SLF, 1993; Meister, 1995), which was largely based on the wording





**Table 1.** European avalanche danger scale (EAWS, 2023c).

| Danger level | Snowpack stability | Likelihood of triggering |
|---|---|---|
| 1-Low | The snowpack is well bonded and stable in general. | Triggering is generally possible only from high additional loads[**] in isolated areas of very steep, extreme terrain[**]. Only small and medium-sized natural avalanches are possible. |
| 2-Moderate | The snowpack is only moderately well bonded on some steep slopes[*]; otherwise well bonded in general. | Triggering is possible primarily from high additional loads[**], particularly on the indicated steep slopes[*]. Very large natural avalanches are unlikely. |
| 3-Considerable | The snowpack is moderately to poorly bonded on many steep slopes[*]. | Triggering is possible even from low additional loads[**] particularly on the indicated steep slopes[*]. In certain situations some large, in isolated cases very large natural avalanches are possible. |
| 4-High | The snowpack is poorly bonded on most steep slopes[*]. | Triggering is likely even by low additional loads[**] on many steep slopes[*]. In some cases, numerous large and often very large natural avalanches can be expected. |
| 5-Very High | The snowpack is poorly bonded and largely unstable in general. | Numerous very large and often extremely large natural avalanches can be expected, even in moderately steep terrain[*]. |

[*] The avalanche-prone locations are described in greater detail in the avalanche forecast (elevation, slope aspect, type of terrain): moderately steep terrain: slopes shallower than about 30 degrees; steep slopes: slopes steeper than about 30 degrees; very steep, extreme terrain: particularly adverse terrain related to slope angle (more than about 40 degrees), terrain profile, proximity to ridge, smoothness of underlying ground surface.

[**] Additional loads: low: individual skier / snowboarder, riding softly, not falling; snowshoer; group with good spacing (minimum 10 m) keeping distances. high: two or more skiers / snowboarders etc. without good spacing (or without intervals); snowmachine; explosives. natural: without human influence.

and definitions used in Switzerland (Föhn, 1985). This adoption of a standardized danger scale marked a pivotal moment for international avalanche warning services, simplifying procedures for all parties involved, and facilitating communication of avalanche danger particularly for forecast users when travelling to different countries (Meister, 1995). Despite for minor changes in 1994, the EADS has been unchanged as of today, not only providing a common way of expressing the avalanche

danger level across institutions and borders, but impacting «the forecasting process itself, as all forecasters are working to an agreed, common, and at least nominally binding definition of avalanche hazard.» (Techel et al., 2018, p. 2698).

The EADS uses two columns to describe each danger level (Table 1). The first column describes snowpack stability and includes a qualitative indication of the frequency of the respective locations. The second column describes the likelihood of triggering an avalanche by indicating the typical avalanche size and their distributions, the likelihood of natural avalanches




occurring or the typical load required to trigger an avalanche. Frequency of avalanches and potential triggering locations or the likelihood of avalanche release are again described qualitatively.

There are several shortcomings with regard to using the EADS as a tool to summarize avalanche conditions in a region:

- The terminology in the EADS is vague, leaving ample room for interpretation. For instance, clear definitions for classes describing snowpack stability and the frequency of triggering locations are lacking.

- Qualitative terms expressing probability or uncertainty are not defined, which according to Morgan (2017) is inadequate as the same term can have different meaning to different people, but also to the same person in a different context. Not surprisingly, even among avalanche professionals large differences in numeric estimates of probability were observed (Thumlert et al., 2020).

- The load necessary to trigger an avalanche is inversely related to snowpack stability (Schweizer and Camponovo, 2001). Thus, both columns in the EADS contain similar and redundant information on *snowpack stability* and triggering.

- The short descriptions of each danger level do not cover the range of all possible combinations. For instance, snowpack stability decreases from *moderately well bonded* to *moderately to poorly bonded* from 2-Moderate to 3-Considerable while its spatial distribution increases from *some* to *many* steep slopes. But the EADS does not provide guidance when the situation is best described by a snowpack that is *moderately to poorly bonded* on *some* steep slopes.

- When the EADS was translated into other languages, sometimes deviations from the original (German) text were introduced. Moreover, it is possible that individual warning services have developed their own guidelines on how to interpret the danger levels over the years, which may be one source for the observed differences in the use of the danger levels in the European Alps (Techel et al., 2018).

Due to these short-comings, a revision of the EADS is required. Such a revision should not only address these points, but must be congruent with the terminology and definitions currently used by EAWS to describe avalanche danger. In addition, a revised EADS must be connected to the forecasting workflow, and the EAWS Matrix.

## 2.2 Look-up tables, workflow and definitions for avalanche danger level assessment

In order to harmonize the use of the danger levels among European avalanche forecasters, the EAWS adopted a look-up table developed by Bavarian forecasters in 2005, the so-called *Bavarian Matrix* (BM; shown in the Appendix Figure A1). The BM was split into two sub-matrices: one relating to the potential for human-triggered avalanches and the other to natural avalanche occurrence. Relying on the terminology of the EADS, a danger level was indicated for each possible combination describing the *probability of avalanche release* and the *distribution of hazardous sites* within the two sub-matrices. The main benefit of the BM was that it provided a suggestion for scenarios for which the EADS provided no guidance. However, the BM inherited the short-comings noted for the EADS as the factors determining avalanche danger, like spatial distribution, avalanche size and probability, were still not clearly separated nor defined.


With the aim to bridge the gap between the cause of snow instability encountered in avalanche terrain and actual risk management decisions, typical situations of avalanche character and snow instability were summarized (e.g., Atkins, 2004; Harvey and Nigg, 2009) in what was later on referred to as *avalanche problem types* in North America or *avalanche problems* in Europe (Statham et al., 2018; EAWS, 2023b). Today, these concepts have become an integral part of regional avalanche forecasts, even though classification schemes differ between North America and Europe.

With some variations, the EADS was adopted in North America in 1994 (Statham et al., 2010). It was used until 2007, when a revised danger scale, the *North American Avalanche Danger Scale* (NADS), was introduced (Statham et al., 2010). This revision also triggered work on a general concept for avalanche hazard assessment resulting in the *Conceptual Model of Avalanche Hazard* (CMAH, Statham et al., 2018). The CMAH identifies the key components of avalanche hazard and structures them into a systematic, consistent workflow for hazard assessments. The method is applicable to all types of avalanche forecasting operations, and the underlying principles can be applied at any scale in space or time (Statham et al., 2018). The workflow sequentially addresses the four questions: «What type of avalanche problem(s) exists? Where are these problems located in the terrain? How likely is it that an avalanche will occur? and How big will the avalanche be?» (Statham et al., 2018, p. 671). While the CMAH has become the standard workflow for avalanche forecasting in North America, it was comparably slowly adopted in regional avalanche forecasting in Europe despite there being a general agreement with the concept. Potential reasons for this slow uptake likely include: (i) The CMAH does (willingly) not conclude with a danger level (Statham et al., 2018). (ii) The CMAH described the locations and spatial distribution of the avalanche problem rather than solely assessing snowpack stability. Analyses in Europe clearly distinguished between the frequency of points with a certain snowpack stability (potential triggering spots) and their actual location (e.g., close to ridge lines, in bowls, ...) (Schweizer et al., 2020; Techel et al., 2020a; Hutter et al., 2021) stating that only the frequency component is relevant for determining the danger level. And lastly, (iii) while the terminology used in the CMAH worked well in the English language, it worked poorly in many European languages (Müller et al., 2016).

In 2016, Müller et al. (2016) attempted to bridge the gap between the concepts introduced in the CMAH and the structure of the Bavarian Matrix leading to the proposition of the *Avalanche Danger Assessment Matrix* (ADAM; see also Figure A2 in Appendix). ADAM provided a workflow similar to the one suggested by the CMAH and integrated the concept of the spatial distribution in the assessment process. ADAM avoided the issue of the poorly defined probability terms used in the EADS by first evaluating snowpack stability against its spatial distribution separately, resulting in a likelihood-score ranging from *unlikely* to *very likely* when merging them. In a further step, likelihood is combined with avalanche size resulting in a danger level. ADAM was presented in two versions, one using the terminology in line with EADS and another one using the terminology from the CMAH. Thus, ADAM provided a first translation between the EADS and CMAH.

At about the time when Müller et al. (2016) developed ADAM, a working group of EAWS presented an updated version of the BM in 2017, which we refer to as *EAWS-Matrix-v2017*. This matrix introduced *avalanche size* as a separate dimension, and, thus, allowed to adjust the danger level described by the *distribution of hazardous sites* and the *probability of avalanche release*. However, most identified shortcoming of the EADS and BM were still present. In the following years, avalanche forecasters in




Europe did not use a common matrix when assigning a danger level; instead each warning service had a preference for one of the three matrices (BM, EAWS-Matrix-v2017, ADAM).

In North America, Thumlert et al. (2020) proposed numerical values to five likelihood terms, which were related to the frequency of natural avalanches releasing in 100 avalanche paths. The five likelihood terms differed compared to any of the other scales in use. Based on the concept presented in ADAM, Thumlert et al. combined these likelihood terms with avalanche

size, introducing a first North American version of an avalanche danger assessment matrix (see also Figure A3 in Appendix).

Common to all these matrices was that they were exclusively based on expert judgments and had been designed by small groups of forecasters (sometimes from only one or two warning services). What was lacking was either data or a consensus within the European avalanche forecaster community on how to resolve the current issues. Consequently, Techel et al. (2020a) tackled this issue and derived a first data-based characterization of the factors determining avalanche danger, which they termed

*snowpack stability*, the *frequency distribution of snowpack stability*, and *avalanche size*. Analyzing a large data set of stability tests and avalanche observations from Switzerland and Norway, Techel et al. showed that the frequency of the locations with the poorest snowpack stability increased with increasing danger level. However, a similarly clear correlation between avalanche size and danger level was not evident. It was observed that the size of the largest avalanche per day and warning region increased only for the higher danger levels. Building upon these insights and drawing inspiration from the matrix layout employed in

ADAM, Techel et al. introduced a data-driven matrix. This new matrix utilized simulated stability distributions along with information on the largest avalanche size (refer to Figure A4 in the Appendix).

Following these developments, a working group of the EAWS adopted the concept and terminology used in Techel et al. (2020a) for the factors determining avalanche danger, namely *snowpack stability*, the *frequency of snowpack stability*, and *avalanche size*, and provided definitions for these factors and their respective classes (EAWS, 2022b). Here, we briefly repeat

these definitions, which are taken from EAWS (2022b):

– *Snowpack stability* is a local property of the snowpack describing the propensity of a snow-covered slope to avalanche (Reuter and Schweizer, 2018). Snowpack stability is described using four classes (Table 2).

– The *frequency distribution of snowpack stability* describes the percentages of points for each stability class relative to all points in avalanche terrain. Thus, the frequency $f$ for all points with stability class $i$ ($n_i$) compared to all points ($n$) is

185 $f(i) = n_i/n$. The frequency distribution of snowpack stability is described in four classes (Table 3).

– *Avalanche size* describes the destructive potential of avalanches (Table 4).

The workflow adopted by EAWS involves decomposing the task of avalanche forecasting into smaller components. In general, the decomposition of complex tasks into smaller tasks is expected to increase the accuracy of the final estimate (MacGregor, 2001). However, as emphasized by MacGregor (2001, p. 107): «Decomposition should be used only when the

190 estimator can make component estimates more accurately or more confidently than the target estimate.»

The quality of human estimates relies on various factors, including the available data, the assessor's expertise in data interpretation, and the assessor's understanding and application of the discrete categories (Stewart, 2001; Hafner et al., 2023).





**Table 2.** Stability classes, and the type of triggering typically associated with these classes, taken from EAWS (2022b, Table 1).

| Stability class | Description |
| --- | --- |
| Very poor | natural / very easy to trigger |
| Poor | easy to trigger (e.g., a single skier) |
| Fair | difficult to trigger (e.g., explosives) |
| Good | stable conditions |

**Table 3.** Frequency classes of snowpack stability, taken from EAWS (2022b, Table 2).

| Frequency class | Description | Evidence (e.g., observations) |
| --- | --- | --- |
| Many | Points with this stability class are abundant. | Evidence for instability is often easy to find. |
| Some | Points with this stability class are neither many nor a few, but these points typically exist in terrain features with common characteristics (i.e., close to ridgelines, in gullies). | |
| A few | Points with this stability class are rare. While rare, their number is considered relevant for stability assessment. | Evidence for instability is hard to find. |
| None or nearly none | Points with this stability class do not exist, or they are so rare that they are not considered relevant for stability assessment. | |

**Table 4.** Avalanche size classes, taken from EAWS (2022b, Table 3).

| Size class | Label | Destructive potential |
| --- | --- | --- |
| 1 | Small | Unlikely to bury a person, except in run out zones with unfavorable terrain features (e.g., terrain traps). |
| 2 | Medium | May bury, injure, or kill a person. |
| 3 | Large | May bury and destroy cars, damage trucks, destroy small buildings and break a few trees. |
| 4 | Very large | May bury and destroy trucks and trains. May destroy fairly large buildings and small areas of forest. |
| 5 | Extreme | May devastate the landscape and has catastrophic destructive potential. |

In our case, these categories pertain to the classes describing snowpack stability, the frequency of the lowest stability class, and the largest expected avalanche size (Tables 2 - 4). While many avalanche warning services communicate these factors in their forecasts, the quality and consistency in selecting these classes have not been explored until now. Nevertheless, inconsis-





tencies in class assignments among different forecasters inevitably influence and diminish the quality of the resulting forecast
(Murphy, 1993), in our context, the avalanche danger level.





## 3 Updating the EAWS Matrix

The revision of the factors determining avalanche danger by the EAWS in 2022 (Sect. 2) lead to a mismatch compared with the terminology used in the *EAWS-Matrix-v2017*. Therefore, an updated matrix was needed.

Most of the previous matrices (EAWS, 2005, 2017) were developed relying on the joint experience of a small group of forecasters consisting, for instance, for the *Bavarian Matrix* of one forecaster from Austria, Germany, France, Italy, Spain, and Switzerland. Unfortunately, the process on how the avalanche danger levels for individual cells within the matrices were assigned, was not documented. Beside the data-driven matrix developed by Techel et al. (2020a), which relied on Swiss data and the Swiss perspective of interpreting danger levels, there was a general lack of data allowing a quantitative characterization of the danger levels.

Expert elicitation is a particularly suitable approach in cases when suitable data is lacking (e.g., Rowe and Wright, 2001). Therefore, EAWS chose to follow a similar path as for previous matrices by combining multiple expert opinions and drawing on the collective knowledge of avalanche forecasters and their perception of the factors and danger levels. However, unlike prior versions, where a small, though likely representative work group made decisions through group discussions, the survey engaged a larger and more diverse pool of domain experts. Experienced EAWS forecasters were considered possessing the appropriate domain knowledge and therefore considered equally capable of performing this task. This approach was grounded in the principle that the aggregated judgment of several experts tends to be more precise than any single individual's opinion, provided judgments are made independently (e.g., Stewart, 2001). Moreover, by inviting EAWS forecasters to contribute their versions of the matrix using the updated terminology and definitions, a higher acceptance of the new matrix was anticipated.

### 3.1 Matrix survey

The matrix was distributed as a survey during the autumn of 2022 with the following instructions:

1. Assign a danger level for each combination of classes describing snowpack stability, the frequency of snowpack stability, and largest expected avalanche size (Tables 2-4). For instance, assign a danger level to a scenario that could be described as «*Many* locations exist, where *poor* snowpack stability prevails. In case that avalanches release, avalanches can reach up to *size 3*.» , where italicized words describe the classes determining avalanche danger.

   (a) Begin with the most unfavorable stability class (*very poor*), which is typically associated with natural avalanches (Table 2), and assign a danger level to every frequency – avalanche size – combination.

   (b) Next, consider *poor* as the determining stability class. Assume that the frequency of locations with stability class *very poor* is *none or nearly none*, or at most *a few* (Table 3).

   (c) Repeat the process for *fair* stability. When *good* is assessed as the lowest stability class, avalanche danger is low.

2. Indicate a primary (more weight) and secondary danger level (less weight) if uncertain between two danger levels.

3. Leave the cell empty if a combination of factors is implausible or if unsure about the appropriate danger level.




Participants were encouraged to fill in all cells for which they felt confident assigning a danger level, leaving the stability
category *fair* as optional with the aim to increase participation rates. The experts answered the survey typically in a 'cold
state', meaning outside an operational setting with a specific situation at hand.

Following best practices for expert elicitation, we instructed forecasters to complete this task independently of other fore-
casters. Most importantly, the danger levels determined for specific combinations of stability, frequency, and avalanche size
should not be discussed among forecasters until after they had submitted their responses.

To derive the updated matrix, we considered the following sources:

– Working group members provided their version of the matrix at a meeting in 2019, and again in 2022 (N = 5 and 9,
respectively). We employed the test-retest reliability methodology (Ashton, 2000) to evaluate the consistency of their
responses and obtain more reliable estimates. Additionally, the second round served as a pilot study to test the survey
distributed to EAWS forecasters outside our working group.

– We invited avalanche forecasters to participate in our survey via the EAWS mailing list and/or the heads of warning
services, receiving 60 responses in total.

– Whenever available, we incorporated quantitative studies into our analysis (N = 2; Swiss data: Techel et al., 2020a;
Hutter et al., 2021).

In total, we received 76 responses from 12 different European countries (Table 5). By combining these sources, we aimed to
generate a comprehensive and robust pool of opinions reflecting the current state of avalanche danger assessment practices in
Europe.

## 3.2 Analysis of survey responses

In line with best-practice approaches when combining judgments from experts (e.g., Dietrich and Spiekermann, 2023), and
not favoring any one opinion, we opted to calculate the median danger level for each combination of stability, frequency, and
avalanche size. In addition, we checked whether the median danger level was also the danger level proposed by the majority of
respondents. Since respondents could provide both a first and second danger level, we weighted their answers accordingly:

– If a forecaster provided a single danger level, this danger level was weighted with 100.

– If a forecaster provided two danger levels, the first danger level was weighted with 67 and the second with 33.

## 3.3 Survey results

Figure 1 shows the distribution of responses for each combination and danger level. As can be seen, a range of factor com-
binations was used for each danger level. While the survey provides insights into the most typical combinations for each
danger level, there were also some combinations, which were rarely or never selected (blank cells). Examining the most fre-
quently selected combinations confirms that stability, frequency, and size tend to increase with higher danger levels. Notably,





**Table 5.** Distribution of the 76 matrix responses received by country, taken from EAWS (2022b, Table D-1). Forecasters in the Czech Republic, Finland, Iceland, Poland and Slovakia were approached, but did not respond.

| Country | N |
| --- | --- |
| Andorra | 3 |
| Austria | 4 |
| France | 7 |
| Germany | 5 |
| Great Britain | 7 |
| Italy | 18 |
| Norway | 15 |
| Romania | 1 |
| Slovenia | 1 |
| Spain | 5 |
| Switzerland | 8 |
| Sweden | 2 |

the combinations with the highest response rate for each danger level often have secondary choices diagonally above or below
that value. This suggests that two factors can offset each other to qualify for the same danger level. For instance, a higher
probability (frequency of snowpack stability) might be balanced by lower consequences (smaller avalanches).

Based on the rank-ordered danger level responses for each cell, we derived the median $D$ and any second $D$ (shown in
brackets) falling within the interquartile range for each combination of stability, frequency, and avalanche size (Figure 2a). We
refer to these two danger levels as $D^1$ and $D^2$, respectively. Analyzing the responses across the 45 cells, we find that 27 cells
contain a $D^2$, indicating considerable variability in opinions.

Examining the proportion of responses aligning with the median $D$ (Figure 2b) reveals strong agreement for cells in the upper
left and lower right corners (proportion $\geq 0.97$), corresponding to the extreme ends of the stability, frequency and avalanche
size spectrum. However, seven combinations, primarily associated with size 1 or size 4 avalanches, show lower agreement rates
(proportion $\leq 0.55$), emphasizing the uncertainty in describing these cells (e.g. *fair-a few-4*).

Figure 2c illustrates the support, or the percentage of responses, for each specific combination. On average, respondents
provided danger level values for 85% of the possible 45 combinations. Notably, cells with *very poor* and *poor* stability received
responses from 72 of the 76 respondents ($\geq 95\%$) for 17 of the 30 combinations. Although stability *fair* was optional in our
survey, it received responses in over 82% of cases when combined with frequency classes *a few* and *some* and avalanche
*sizes 1-3*. *Fair* stability had lower response rates when paired with avalanche size 4 ($\leq 66\%$) or size 5 ($\leq 50\%$). Possibly, this
indicates that a considerable share of forecasters rated these combinations as less plausible.


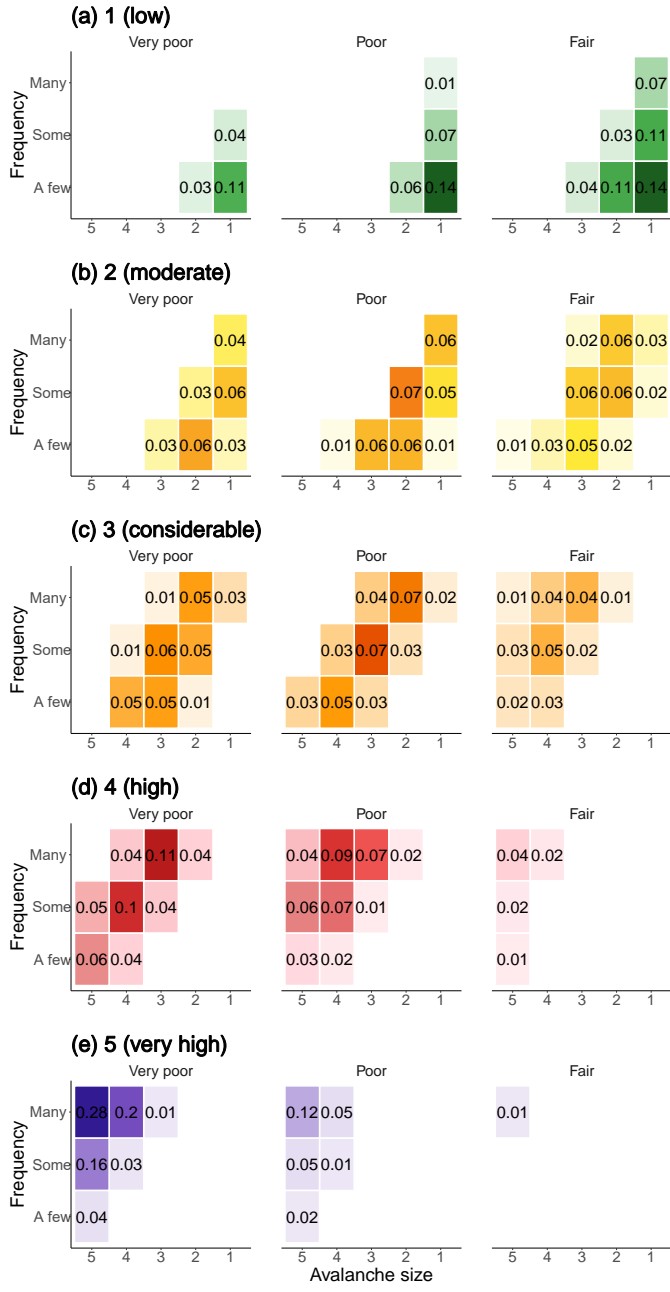

**Figure 1.** Distribution of survey responses for each danger level (a: $D = 1$ to e: $D = 5$). Shown are the proportions for each combination of stability, frequency, and avalanche size summing up to 1 for each $D$. Values are displayed if they received $\geq 0.01$ of the votes. Stronger color saturation indicates a larger proportion of responses favoring a specific combination.



**Figure 2.** Survey responses for each combination of stability (panels), frequency (y-axis), and avalanche size (x-axis). The median danger level ($D^1$) is displayed in a), while the proportion of responses agreeing with $D^1$ is shown in b). The proportion of responses providing a danger level estimate is depicted i c). Stronger color saturation indicates a larger proportion of responses.





## 4   The updated EAWS matrix

The findings presented in Section 3 led to the development of an updated matrix (Figure 3), hereafter referred to as EAWS Matrix, which was officially accepted by the EAWS General Assembly in June 2022. The updated matrix provides a comprehensive framework for assessing avalanche danger, taking into account three factors to determine avalanche danger: snowpack
stability (shown as panels in Figure 3), frequency (along the y-axis), and avalanche size (along the x-axis).

The design of the EAWS Matrix is based on the recognition that the frequency of locations with the weakest snowpack stability is often decisive for determining the avalanche danger level (Techel et al., 2020a). Therefore, the matrix is structured to address the three lowest stability classes *very poor*, *poor*, and *fair*. For each stability class, combinations of frequency and avalanche size are summarized in separate panels (Figure 3). When using this matrix to assess danger levels, forecasters follow
a systematic approach, starting from left to right. They begin by considering the lowest stability class. If this class corresponds to a frequency of *none or nearly none*, they proceed to the next stability class on the right, and so forth, indicated by the arrows in Figure 3. When the snowpack stability is evaluated as *good*, the danger level is automatically assigned as *1-Low*, regardless of the values of the other two factors. It is important to note that while stability and frequency are closely linked, the assessment of avalanche size is independent. Forecasters select the largest size likely given the observed or anticipated conditions.

To accommodate forecasters sometimes providing differing danger level estimates for specific combinations of stability, frequency, and size (Figure 2b), the matrix displays either one or two danger levels: the median danger level, referred to as $D^1$, using the integer value of the danger level (i.e. 1 for *1-Low*). $D^1$ also defines the color of a cell. In addition, a second danger level ($D^2$) is shown in brackets, if the interquartile range of the danger level responses included a second danger level, which was different from $D^1$. Cells without coloring represent instances where fewer than 70% of respondents provided a danger
level estimate (Figure 2c).

To facilitate the application of the EAWS Matrix, the EAWS working group has developed a workflow that outlines the necessary steps for determining the avalanche danger level within a warning region (Müller et al., 2023). The workflow involves assessing all relevant avalanche problems, evaluating their snowpack stability, frequency, and avalanche size, and then using the EAWS Matrix to assign a danger level to each problem. The highest resulting danger level among the considered
avalanche problems is communicated for the given warning region. This structured approach ensures that all relevant factors are considered, aiming for a more consistent evaluation of avalanche danger.

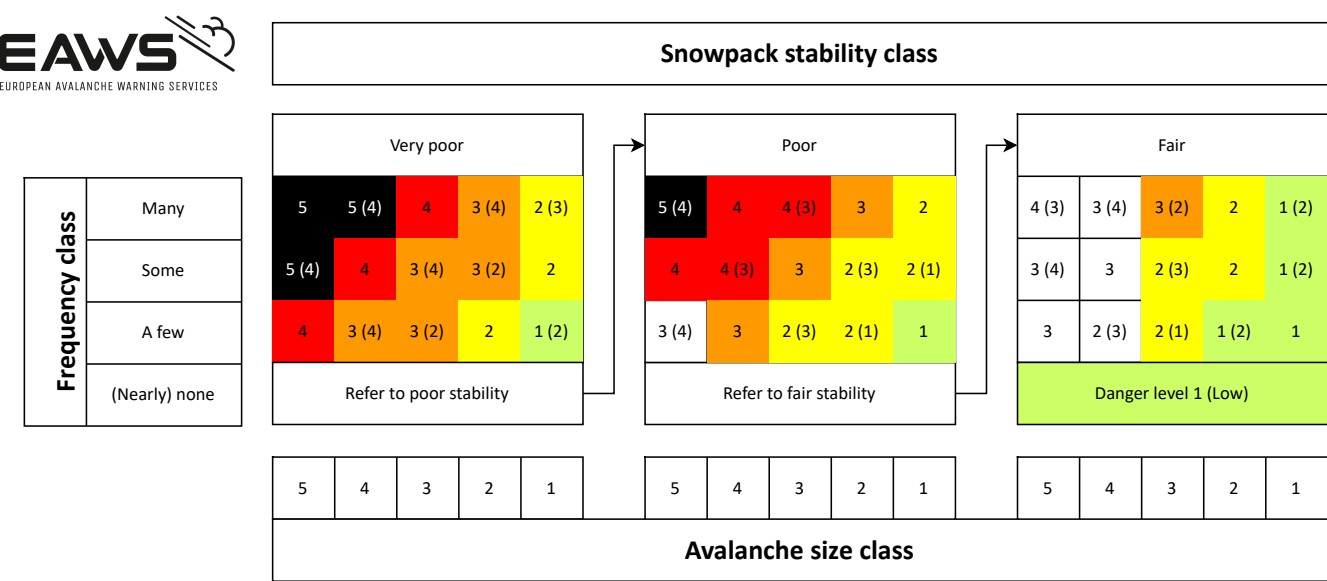

**Figure 3.** EAWS Matrix, as accepted by the EAWS General Assembly in 2022 (taken from EAWS, 2022b), is organized into three distinct panels, each corresponding to one of the three lowest snowpack stability classes. Within each panel, the frequency (on the y-axis) is plotted against avalanche size (on the x-axis). To navigate through the matrix, arrows interlinking the panels indicate to transition to the next stability class, specifically when the frequency is evaluated as *none or nearly none*. According to the workflow (Müller et al., 2023), forecasters first assess the frequency of the most vulnerable locations, progressing from left to right across the panels. Upon identifying a relevant stability-frequency combination, forecasters then evaluate the corresponding avalanche size, culminating in a recommended danger level.




**Table 6.** Use of matrix during winter 2022/2023. Warning services are aggregated to groups of services using the same workflow and forecasting software or when spatially continuous.

| | country (regions) | N cases (n days) | use | Factors published | Factors → $D$ | Danger level $D$ in forecast software | deviations from $D^1$ |
|---|---|---|---|---|---|---|---|
| ATB | **B**avaria) Germany (Austria (except Tyrol) | 2347 (155) | operat'l | no | direct link | $D^1$ suggested, $D^2$ shown | allowed (5.7%) |
| CAT | Spain (**Cat**alunya) | 289 (56) | operat'l | yes | direct link | only $D^1$ indicated | (0%) |
| NOR | **Nor**way | 7014 (181) | operat'l | yes | direct link | only $D^1$ indicated | allowed (16%) |
| SWE | **Swe**den | 671 (146) | operat'l | yes | direct link | $D^1$ and $D^2$ shown | allowed (9.7%) |
| SWI | **Swi**tzerland | 595 (81)* | test (live) | no | no link | matrix not used | (47%) |
| TST | Austria, Italy (**T**yrol, **S**outh Tyrol, **T**rento) | 1020 (142) | operat'l | yes | direct link | only $D^1$ is shown | allowed (2.0%) |
| VAR | Spain (**V**al d'**Ar**an) | 216 (129) | operat'l | yes | direct link | only $D^1$ indicated | allowed (6.5%) |

Reading example: For warning services in ATB, forecasts were available for 155 forecast days, resulting in 2347 unique forecasts, where uniqueness is defined by date, issuing warning service, avalanche problem, combination of factors, and danger level. The EAWS Matrix was used operationally. Factors determining avalanche danger were assessed but not published. In the software tool, there was a direct link between the selected factors and the danger level. $D^1$ was suggested by software, but $D^2$ was also shown. Overruling of $D^1$ was possible and was used 5.7% of the time.

* In Switzerland, the respective forecast for each of the three forecasters on duty is considered.




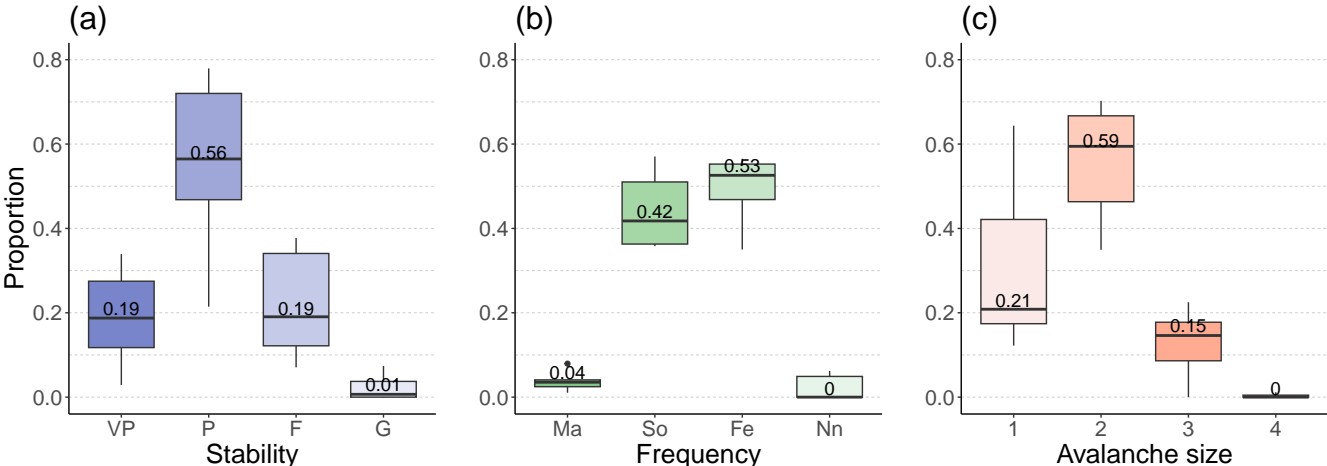

**Figure 4.** Use of classes during winter 2022-2023 describing (a) snowpack stability, (b) frequency, and (c) avalanche size. The boxplots represent the distribution of the class labels summarized for all warning services in Table 6. Abbreviations for i) stability are VP-very poor, P-poor, F-fair, G-good, ii) frequency are Ma-many, So-some, Fe-a few, Nn-none or nearly none, iii) and the avalanche size classes 1-5 (size-5 never used).

## 4.1 Use of EAWS Matrix in winter 2022-2023

EAWS members were encouraged to apply the definitions, workflow and updated EAWS Matrix in their operations during the winter season 2022-2023. In total, 15 EAWS avalanche warning services logged their choices for the three factors during

operational forecasting, either for the entire season or for extended periods. For the purpose of this analysis, we aggregated warning services to groups of services using the same workflow and forecasting software, and when spatially continuous. Table 6 provides an overview. In most of these warning services, workflow and matrix were implemented in the operational workflow and forecasting software. An exception was the Swiss warning service (SWI), where the objective was to first gain insights on the reliability of estimating the factors describing avalanche danger according to the partly revised definitions (Tables 2-4). In

SWI, forecasters were advised not to use the EAWS Matrix.

Analyzing each of the three factors determining avalanche danger separately reveals several interesting findings. Firstly, snow stability was most commonly rated as *poor* (P), with a median proportion across the warning services of 0.56 (Figure 4a). The terms *a few* and *some* were the predominant choices for frequency, with proportions of 0.53 and 0.42, respectively (Figure 4b). Avalanche size also displayed a clear dominance of *size-2* with a median of 0.59 of all responses (Figure 4c).

As can be noted in the range of values displayed in the boxplots in Figure 4, the use of the classes varied quite strongly between warning services. However, we refrain from interpreting these variations, as a multitude of factors, including seasonal and snow-climatological differences, data collection methodologies, and potential differences in the interpretation of class definitions, may all contribute to the observed differences.





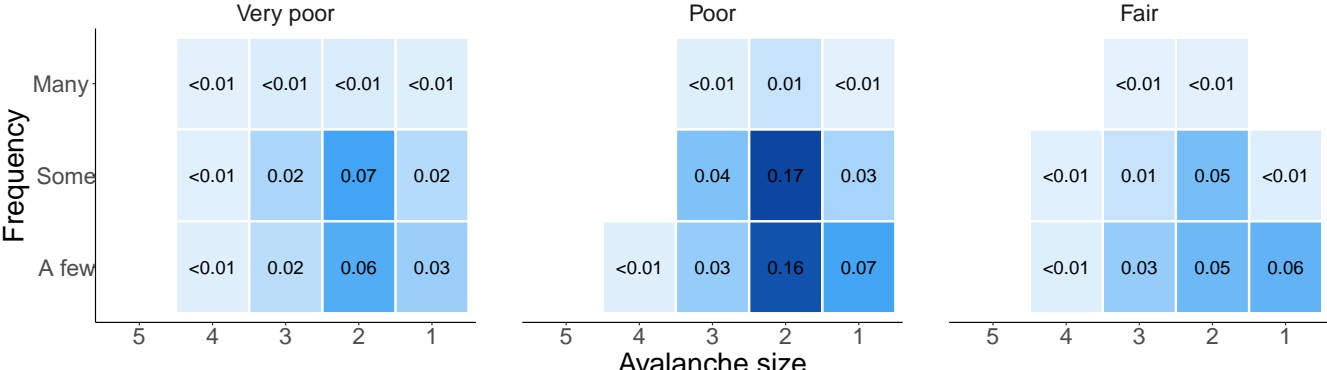

**Figure 5.** Use of individual matrix cells. The values represent the unweighted means of the respective proportions for all groups of warning services as in Table 6. Stronger color saturation indicates a larger proportion of responses.

Examining the use of specific combinations of these factors (Fig. 5), we observe that the two single most frequent combi-
nations were *poor-some-2* and *poor-a few-2*, representing 17% and 16% of the responses. Combinations featuring *very poor*
or *fair* stability were only used about half of the time compared to *poor*. Combinations denoting *many* for frequency were
seldom used, with proportions falling below or equal to 1% . Furthermore, combinations featuring avalanches of *size 4* were
exceptionally rare, while *size 5* was virtually never selected, underscoring their association with extremely rare and unusual
avalanche conditions.

Analyzing differences between $D$ given by a forecaster for a specific combination of stability, frequency, and size, and $D^1$,
as indicated in the Matrix, shows that almost all warning services exhibited some level of deviation from the Matrix during the
winter occasionally (Tab. 6). With 47% of the cases, deviations were most frequent in SWI, where the Matrix was not consulted.
In NOR, where the Matrix was used operationally, deviations were also observed comparably often (16%). In contrast, CAT
never deviated from $D^1$, and in TST the proportion of deviations was also small (2%).

Figure 6 shows the agreement between $D$ and $D^1$ for all combinations of factors and the seven groups of warning services,
as defined in Table 6. The left column shows the proportions of cases when $D = D^1$, and the right column where $D \neq D^1$
for each group of warning services. Of interest are primarily the combinations for which high proportions of $D \neq D^1$ were
observed. Not surprisingly, these were most frequent for SWI and NOR, followed by SWE and VAR. $D \neq D^1$ was $> 0.5$
in eight cases (SWI 6, NOR 2), being most pronounced for *poor-many-3* with 100% deviation for SWI and 75% for NOR.
NOR had comparably low proportions of $D = D^1$ for *fair*, while in SWI a variety of cells showed deviations, with no obvious
pattern. For the most frequently used combination, *poor-some-2*, all services (except CAT) deviated from the Matrix in at least
4% of cases. Disagreement with $D^1$ was particularly large in SWI, where $D^2 = 3$ was chosen in 53% of cases. The pattern
of optioning for $D^2 = 3$ was also comparably evident in SWE, NOR, and VAR (14-19% of cases). As several forecasters
assessed each situation in SWI, a majority opinion can be derived for each case. Considering the 40 occasions when the factor
combination using majority voting resulted in *poor-some-2*, the tendency to deviate from $D^1$ was even more pronounced: the





majority-voted $D$ was $D^2 = 3$ in 25 cases (63%) and $D^1 = 2$ in 12 cases; in three cases it was undecided between the two danger level options.





**Figure 6.** Proportion of cases that $D = D^1$ or $D \neq D^1$ were used by groups of warning services (center column). Abbreviations according to Table 6. Cells are displayed if the number of cases was $\geq 5$.

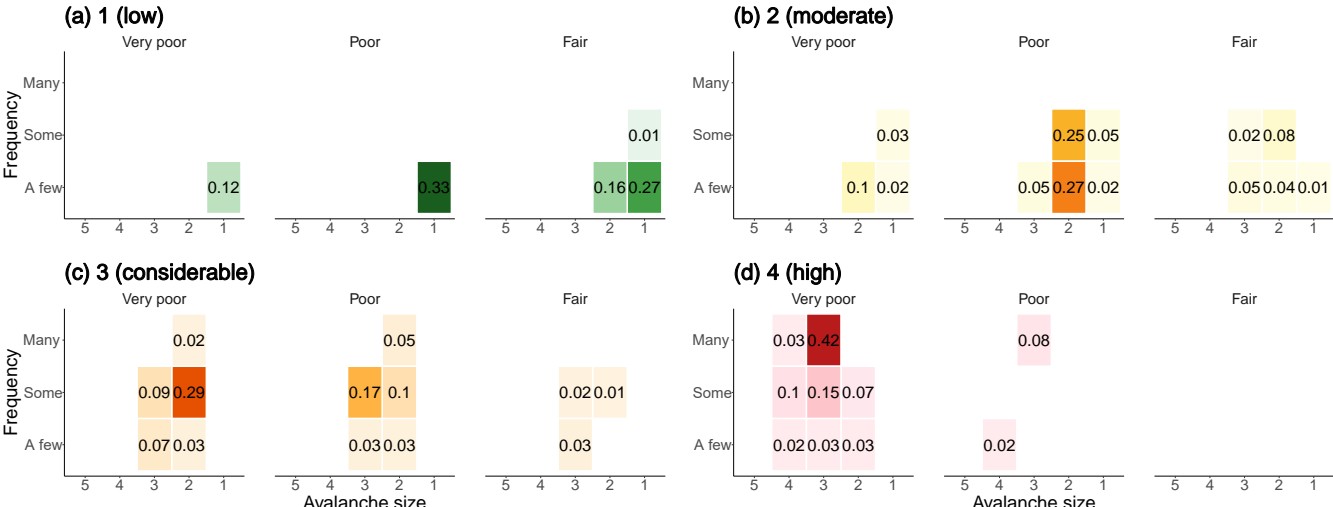

**Figure 7.** Proportion of cases that a specific danger level (a: $D = 1$ to d: $D = 4$) was used for a specific matrix combination. Proportions are derived by summing up all cases $n$ across warning region groups. Values are shown for all cells with a proportion $\geq 0.01$. Colour shading correspond to the proportion of cases.

The spread of cells used for each $D$ is shown in Figure 7. $D = 1$ ($n = 2203$, 18% of all cases) is closely linked to avalanche size 1 and *a few* locations with predominantly *fair* or *poor* stability. In addition, $D = 1$ is associated with *good* stability in 10%
cases (*good* stability is not shown in Fig. 7). $D = 2$ ($n = 6806$, 56%) has its center of gravity at *poor* stability with *a few* or *some* locations and avalanche size 2. $D = 3$ ($n = 3025$, 25%) is scattered around the combinations *very poor - some - size 2* and *poor - some - size 3*, while $D = 4$ ($n = 118$, 1%) is clearly centered at *very poor - many - size 3*.

In Figure 8 we show the typical factor combinations for each group of warning services. Overall, the most commonly used descriptions of $D$ exhibited strong similarities, whether we examined the factors independently or in combination. Never-
theless, discernible distinctions emerge. For instance, in NOR and SWE, $D = 1$ were most frequently associated with *poor* stability, whereas elsewhere *fair* stability dominated. Similarly, at $D = 3$ in NOR, CAT, and to some extent in ATB, there was a greater prevalence of *very poor* stability compared to other warning services. Frequency terms employed to describe danger levels generally exhibited similarities, with the exception being NOR, where frequency for $D = 4$ was more frequently described as *some* rather than *many*, often in combination with larger avalanches. For example, at $D = 4$, avalanche size in
NOR ranged between size 3 and 4, while in SWI, it typically varied between size 2 and 3. However, it's worth noting that the days of $D = 4$ were relatively rare (118 cases), suggesting that these results may not be fully representative. Avalanche size exhibited greater variation at higher danger levels but demonstrated remarkable uniformity at $D = 1$ and $D = 2$. And lastly, in SWI the most frequently used combination for $D = 3$ (*poor-some-2*) was the combination, which was the most frequent for $D = 2$ in many other warning services (CAT, SWE, TST, VAR). Nevertheless, when assigning equal weight to the various
datasets, distinct patterns emerge regarding the most prevalent combinations (see Figure 8).





| | | Stability | | | | | Frequency | | | | | Avalanche size | | | | | | |
|---|---|---|---|---|---|---|---|---|---|---|---|---|---|---|---|---|---|---|
| | | **Danger level** | | | | | **Danger level** | | | | | **Danger level** | | | | | | |
| | | 1 | 2 | 3 | 4 | 5 | 1 | 2 | 3 | 4 | 5 | 1 | 2 | 3 | 4 | 5 | | |
| **EADS** | | F | F | P (VP) | VP (P) | VP | Fe | So | So, Ma | Ma | Ma | 1/2 | n.d., <4 | ≤3/4 | 3, 4 | 4/5 | | |
| **EAWS Matrix** | individual | F | F (P) | P (VP) | VP (P) | VP | Fe | Fe (So) | So (Ma) | Ma (So) | Ma | 1 | 2 (1) | 3 | 4 (5) | 5 | **Stability** | |
| **(v2022)** | combined | F | P | P | VP | VP | Fe | So | So | Ma | Ma | 1 | 2 | 3 | 3 | 5 | very poor | VP |
| **ATB** | individual | F | P | VP (P) | VP | - | Fe | Fe (So) | So | Ma | - | 1 | 2 | 3 (2) | 3 | - | poor | P |
| | combined | F | P | P | VP | - | Fe | Fe | So | Ma | - | 1 | 2 | 3 | 3 | - | fair | F |
| **CAT** | individual | F | P | VP | - | - | Fe | So | So | - | - | 1 | 2 | 2 | - | - | good | G |
| | combined | F | P | VP | - | - | Fe | So | So | - | - | 1 | 2 | 2 | - | - | **Frequency** | |
| **NOR** | individual | P | P | VP (P) | VP | - | Fe | Fe (So) | So (Fe) | So (Ma) | - | 1 | 2 | 2 (3) | 3 (4) | - | many | Ma |
| | combined | P | P | VP | VP | - | Fe | Fe | So | So | - | 1 | 2 | 2 | 4 | - | some | So |
| **SWE** | individual | P (F) | P | P | - | - | Fe | Fe (So) | So | - | - | 1 (2) | 2 | 3 (2) | - | - | a few | Fe |
| | combined | P | P | P | - | - | Fe | So | So | - | - | 1 | 2 | 3 | - | - | (nearly) none | Nn |
| **SWI** | individual | F | P | P | VP | - | Fe | Fe (So) | So | Ma | - | 1 | 2 | 2 (3) | 2 (3) | - | **Avalanche size** | |
| | combined | F | P | P | VP | - | Fe | Fe | So | Ma | - | 1 | 2 | 2 | 3 | - | extremely large | 5 |
| **TST** | individual | F | P | P (VP) | VP | - | Fe | So (Fe) | So (Ma) | Ma | - | 1 | 2 | 2 (3) | 3 | - | very large | 4 |
| | combined | F | P | P | VP | - | Fe | So | So | Ma | - | 1 | 2 | 3 | 3 | - | large | 3 |
| **VAR** | individual | F | P | P | - | - | Fe | Fe (So) | So | - | - | 1 | 2 | 2 (3) | - | - | medium | 2 |
| | combined | P | P | P | - | - | Fe | So | So | - | - | 1 | 2 | 3 | - | - | small | 1 |

**Figure 8.** Typical descriptions of $D$ based on the European Avalanche Danger Scale (EADS), the updated matrix (Figure 4), and their frequency of use during the winter season of 2022/2023. In the "individual" rows, we have tallied the classes describing the factors independently, irrespective of their combination with others. These rows display the most commonly occurring class, and when a second class is present in at least 30% of cases, it is enclosed in brackets. Conversely, in the "combined" rows, we showcase the classes from the most prevalent combination of stability, frequency, and size. For reference, "n.d." denotes instances where no specific class was defined.

## 4.2 Consistency in assessing factors determining avalanche danger

To investigate potential inconsistencies in assigning classes to the three determining factors, two warning services, NOR and SWI, conducted tests during the winter season 2022-2023. In SWI, three forecasters made independent assessments of the classes on a daily basis, but only for a subset of the Swiss forecasting domain, resulting in 564 pairwise comparisons (219 unique cases on 73 forecast days). In NOR, on three separate occasions, between 6 and 12 forecasters independently conducted comprehensive avalanche danger assessments for one selected region for the following day, resulting in 117 pairwise comparisons. As mentioned before, forecasters in SWI were explicitly instructed not to consider the Matrix for assigning $D$ in order to mitigate any influence the Matrix might have on their class choices. Conversely, in NOR, incorporating the Matrix is a standard part of the forecaster workflow when assigning a danger level.

We consider the mean agreement between any two forecasters assessing the same forecasting scenario as a measure of consistency among forecasters. Here, agreement means the selection of the same class for a specific factor. With this approach, we obtain an estimate of how reliably two randomly chosen forecasters would obtain the same class. In SWI, the pairwise agreement rate for selecting the same class describing the factors was 59% for stability, 63% for frequency, and 74% for avalanche size (Table 7). The agreement rate for the combination of all three factors was 30%, while in 4% of cases, the classes for all three factors differed. In NOR, the classes are set for each avalanche problem where the main problem is said to be decisive for the danger level. Therefore, we conducted the analysis for the avalanche problem considered decisive but also for



**Table 7.** Pairwise agreement when estimating the factors describing avalanche danger. $n$ indicates the number of pairwise comparisons, $N$ the number of unique cases (unique day and warning region).

| Source | Avalanche problem | Stability | Frequency | Size | All the same | All different | n (N) | #Forecasters |
|---|---|---|---|---|---|---|---|---|
| NOR | decisive | 49% | 50% | 77% | 21% | 3% | 117 (3) | 6, 8, 12 |
| | same | 54% | 51% | 73% | 21% | 7% | 169 (3) | 6, 8, 12 |
| SWI | decisive | 59% | 63% | 74% | 30% | 4% | 564 (219) | 2 or 3 |

cases where they were given in a different order (values in brackets). The respective pairwise agreement rates were 49% (54%) for stability, 50% (51%) for frequency, and 77% (73%) for avalanche size (Table 7). All three factors were the same in 21% (21%) of cases, while in 3% (7%) of cases, the values on all three axes differed. It is of note, however, that the most frequently

chosen classes were *poor* for stability (67%), *some* for frequency (54%), and *size-2* for avalanche size (73%). Therefore, simply using the most typical value would already yield a relatively high agreement rate.





## 5 Discussion

### 5.1 Matrix survey and updated EAWS Matrix

A new EAWS matrix was needed to be congruent with the revised definitions of the factors determining avalanche danger and the workflow for assessing regional avalanche danger (Müller et al., 2023, summarized in Sect. 2). Due to a lack of objective data, expert elicitation was conducted by asking European avalanche forecasters in a survey to assign a danger level ($D$) to each combination of factors (Section 3). This task required survey participants to understand, interpret and apply the partly revised, purely descriptive definitions of the classes for snowpack stability, frequency, and avalanche size (Tables 2-4), and link them to the danger levels.

It is well known that the same word can have different meaning to different people depending on their background, culture or language, and that this meaning may also differ to the same person in a different context (e.g., Ogden and Richards, 1925; Morgan, 2017). Thus, variations in the combination of factors and corresponding danger level had to be expected. However, to reduce the influence of a specific language or cultural background on the final matrix, forecasters from throughout EAWS were approached. It is, therefore, not surprising that the responses obtained from the 76 respondents revealed considerable variability in the assignment of $D$ across most factor combinations (refer to Section 3). Moreover, as shown in EAWS (2022a), 'cultural' differences can be noted when comparing responses by country, with, for instance, the mean response by Scottish forecasters resulting in five matrix cells with $D^1 = 5$, while only two of the matrix cells were assigned $D^1 = 5$ by Norwegian or Swiss forecasters. Such 'cultural' differences have been noted previously, as, for instance, when assigning danger levels (Lazar et al., 2016; Techel et al., 2018) or when estimating avalanche size (Hafner et al., 2023). Despite the EADS being in use for three decades, the absence of unambiguous, standardized guidelines and a common understanding and interpretation of the definitions across European Avalanche Warning Services likely contributes to these variations (Techel et al., 2018).

The revised EAWS Matrix presented in Section 3.1 (Figure 3) is based on 'consensus', which in our case is defined as $\geq 75\%$ of responses suggesting one danger level and means that a clear majority of respondents favored one specific danger level. Such consensus existed for only 18 of the 45 possible factor combinations (Fig. 2b). These cells of comparably high agreement are scattered across the matrix, with no obvious pattern connected to one of the factors or to $D$. In the remaining 27 cases, up to 50% of respondents suggested an alternative danger level, shown in brackets in the EAWS Matrix and denoted as $D^2$ (Figure 3). Not surprisingly, the cells with highest agreement define the limits of the danger scale (*fair-a few-1*, $D^1 = 1$: 97%; *very poor-many-5*, $D^1 = 5$: 99%; Figure 2b). The other two cells that stand out with regard to a high agreement of responses are *very poor-many-3* ($D^1 = 4$: 85%) and *poor-some-3* ($D^1 = 3$: 84%). These two combinations align well with the description of danger levels 3 (considerable) and 4 (high) in the EADS (Fig. 1), which likely explains the clear preference for one danger level in the survey.

### 5.2 Matrix usage

The majority of the analyzed warning services integrated the EAWS Matrix into their forecasting software. Typically, the default suggestion presented to forecasters after selecting a factor combination was $D^1$, nudging them towards this option over





$D^2$. While forecasters had the autonomy to override $D^1$, indicating a disagreement with the EAWS Matrix and the 'cold state' represented by the survey, they also had the undesired option to subtly influence the suggested $D^1$ by adjusting one or more factors. This practice could mask a forecaster's true assessment and potentially impact the choice of $D$ in an unintended manner. Although the extent of this nudging effect and the frequency of factor adjustments to achieve a desired $D^1$ remain unclear, it is reasonable to assume that they influenced decisions at least to some extent. Notably, the nudging effect of the EAWS Matrix is

deliberate once a consensus has been reached for all combinations, aiming to enhance consistency in assigning danger levels. However, it's crucial to ensure that this nudging does not compromise the forecasters' discretion regarding input parameters, which may have occurred sporadically.

Moreover, it is noteworthy that feedback obtained from various seminars with forecasters following the winter season indicated that less-experienced forecasters highly appreciate the guidance offered by the EAWS Matrix, potentially also leading

to a greater tendency to agree with $D^1$. In contrast to the general setup of including the EAWS Matrix in the daily workflow, in Switzerland, the focus was primarily on evaluating the reliability of estimating the factors determining avalanche danger, regardless of $D$. Therefore, forecasters were instructed to assess the factors to the best of their knowledge and to ignore the EAWS Matrix to reduce the potential nudging effect of the Matrix during this testing season. Given these different setups and intentions, it is, thus, not surprising that some warning services adhered to $D^1$ most of the time (i.e. ATB, CAT, TST chose $D^1$

$\geq 93\%$), while other services deviated more frequently from $D^1$, with a particularly large share of deviations observed in SWI (46%).

Despite the differences in the use of the Matrix during the forecasting season, which may have somewhat amplified choosing $D^1$, our analysis suggests that forecasters largely concurred with the avalanche danger level proposed by the EAWS Matrix (Figure 6). This observation is supported by the fact that for most Matrix cells there was a tendency to agree with $D^1$ (Fig. 6)

and that for most danger levels a similar set of typical factor combinations resulted (Fig. 8), regardless whether the Matrix was used or not (SWI). This supports the assumption that, overall, the approach of deriving a consensus-based matrix by eliciting expert knowledge resulted in generally applicable factor combinations.

Despite the overall usage patterns aligning well with the danger level descriptions in the EADS (Tab. 1), there are two cases, which need particular attention, namely (1) the shift in stability for $D = 2$ from *fair* (EADS) to *poor* (usage), and (2)

the closeness of the typical factor combinations describing $D = 2$ and $D = 3$, which are the same for two of the three factors (Figure 8).

All warning services described stability most often as *poor* when giving $D = 2$, and, therefore, deviated from the description of stability in the EADS. However, the usage of *poor* stability was compensated by using a lower frequency class *a few* (to *some*) by some warning services (ATB, NOR, SWE, SWI), while CAT and TST generally used *poor* stability in combination

with *some* locations. In other words, the latter two warning service groups kept the frequency class from EADS, but shifted stability to a lower class, which can be interpreted as a true shift from EADS.

Danger levels $D = 2$ and $D = 3$ collectively account for 81% of the data, which is very similar to previous analyses for the European Alps (about 80%; Techel et al., 2018). In the EADS and in the EAWS Matrix, their descriptions primarily differed by stability ($D = 2$: *fair* vs. $D = 3$: *poor*), and - in the Matrix - also by avalanche size ($D = 2$: *size 2* vs. $D = 3$: *size 3*), but





less by frequency. The shift to *poor* stability in the usage data for $D = 2$ reduces the variations between these two danger levels. Moreover, it must be expected that the distribution of cases within $D = 3$, that is the number of cases at the lower and higher end of the level, are unbalanced towards the lower end. While we have no direct evidence for that within our data-set, the distribution of sub-levels in Switzerland, which attempt to capture exactly these variations within a level, showed that a 3- (avalanche conditions considered to be low in the level) was two times more frequent compared to a 3+ (high in the level) when

considering the forecasts from a 4-year period (Techel et al., 2020b), but that these were four times more frequent during the winter 2022-2023 (unpublished data). Assuming similar distribution of avalanche conditions in other regions in the Alps (ATB, TST) indicates that there are many more situations with $D = 3$ close to $D = 2$ compared to cases close to $D = 4$. Moreover, and again this relies only on Swiss data (Techel et al., 2020b), forecasters sometimes seem to be hesitant to lower from $D = 3$ to $D = 2$. In contrast, in Norway, there was a clear preference for *very poor* stability for $D = 3$. We can only speculate whether

this is related to a more balanced distribution of cases within $D = 3$, or whether Norwegian forecasters have a slightly different representation of *very poor* compared to other warning services, or whether this is related to the much larger regions in Norway. However, all of these combinations fall well within the responses obtained in the survey (Fig. 1). In other words, this indicates that the underlying distribution must also be taken into account when considering the most frequently used combination as an approach to obtain a typical description for each danger level. Seen from that perspective, the two approaches - survey and

usage - provide valuable different views to obtain a typical characterization of each danger level.

When examining the EADS, it becomes apparent that the combination *poor-some-2* and *poor-a few-2* encompass elements of the description of both $D = 2$ and $D = 3$. Consequently, it represents one of these intermediate states, which are frequently encountered and utilized by avalanche forecasters, but which are not adequately captured by the EADS. Moreover, the EADS does not assign a specific avalanche size to $D = 2$ and only states that «very large avalanches (*size 4*) are unlikely» (Table

1), basically only excluding *size 5*. In the case of these combinations, the agreement rates in the survey do not significantly differ from the agreement rates observed during operational usage in 2022-2023. Figure 6 illustrates that SWI, in particular, tends to favor $D = 3$ for the combination *poor-some-2*, while NOR, SWE, and VAR also occasionally lean toward assigning $D = 3$ for this specific scenario, the other services and the EAWS Matrix currently favor $D = 2$. As mentioned before, SWI most often deviated from $D^1$, which is likely related to the fact that SWI forecasters generally assign $D$ without consulting the

Matrix. Nonetheless, it is of note that forecasters in SWI generally achieve high agreement on $D$ (Lucas et al., 2023), but poor agreement on the factor classes.

It is potentially problematic that the two most common danger levels strongly intersect and that there is no consensus among forecasters on when to assign them given the current definitions. Obviously, a clear distinction is necessary to improve consistency between forecasters and to ease communication with the public.

**5.3 Estimating the factors determining avalanche danger**

The current EAWS Matrix, along with its predecessors, rely on the concept of decomposing the complex task of assigning a danger level by breaking this task into smaller components. Such decomposition is expected to increase the accuracy of the final estimate if the estimator is able to make more accurate estimates of the individual components (MacGregor, 2001). Moreover,





it is also a logical step to consider all relevant input parameters, which can reliably be estimated in the judgment process. How-
ever, there are numerous factors potentially influencing the ability to make accurate estimates, whether concerning individual
input factors or the danger level as a whole. These include the environmental predictability of the avalanche conditions, the
correspondence between the available data (input) and the avalanche conditions (output), the match between the environment
and the forecaster, the reliability of the forecaster acquiring the relevant information, or the skill of the forecaster processing
this information (e.g., Stewart and Lusk, 1994; Stewart, 2001).

In the matrix survey, the task was detached from a real forecasting situation as the participating forecaster had to assign
combinations of the three factors to danger levels without relating to a specific avalanche situation. Therefore, variations
likely resulted primarily from the fact that each forecaster has a slightly different representation of what the terms describing
snowpack stability, frequency, and avalanche size mean, and how these relate to the danger level. In contrast, during operational
use, all the before-mentioned points come into play, each of which may potentially increase variations and may thus reduce
consistency between forecasters. However, in addition to these points, rather vague and imprecise definitions, some of which
were only introduced prior to this winter season (i.e., the frequency classes; Tab. 3), likely also impacted consistency when
estimating the axis labels.

The comparably low agreement rate between any two forecasters estimating the factors for exactly the same situation (Sect.
4.2) certainly raises questions regarding the reliable estimation of the factors determining avalanche danger, and, consequently,
how this unreliability impacts the assignment of the danger level. To make this clearer, imagine that two forecasters assess
exactly the same avalanche conditions and that their assessments agree for two of the three factors, and differ for one of
the factors by one neighboring class. Assuming that this kind of disagreement is the normal case, and testing this for all
possible combinations in the Matrix, the resulting $D^1$ would differ between the two forecasters in 42% of cases. Assuming
that forecasters become more consistent with continued use, say two forecasters differ on one of the factors 50% of the time
while agreeing in all other situations, the resulting $D^1$ would still differ 21% of the time. As only one of the two forecaster's
estimate can be «correct» and assuming both forecasters being equally competent, such inconsistency in the assessment of
classes inevitably reduces accuracy. In the given example, accuracy would be at most 0.89 - the factor required to achieve an
agreement rate between two forecasters of 0.79, solely due to this inconsistency (Techel, 2020, p. 35).

In Switzerland, forecasters historically did not use the Matrix nor were they constrained to select from a predefined set of
classes to describe the factors determining avalanche danger. Instead, a diverse range of words describing these factors (see also
Hutter et al., 2021) were employed in daily forecaster discussions, where the team collectively determined the danger level.
Therefore, the introduction of a short list of classes for stability and frequency, and forcing forecasters to use these classes, likely
contributed to the low agreement rates (stability: 59%, frequency: 63%). This stands in contrast to Swiss forecasters achieving
significantly higher pairwise agreement rates when selecting an avalanche problem ($\geq 74\%$, $N = 15000$) or the danger level
(86%, $N = 45000$, unpublished data). Interestingly, in Norway, where forecasters have long used this system (Müller et al.,
2016) and where the factors are publicly communicated in avalanche forecasts (Engeset et al., 2018), the agreement rates
were similarly low ($\leq 54\%$). However, the Norwegian numbers are only based on three situations with comparatively dynamic
weather and avalanche conditions, which may have heightened the decision-making complexity for forecasters.





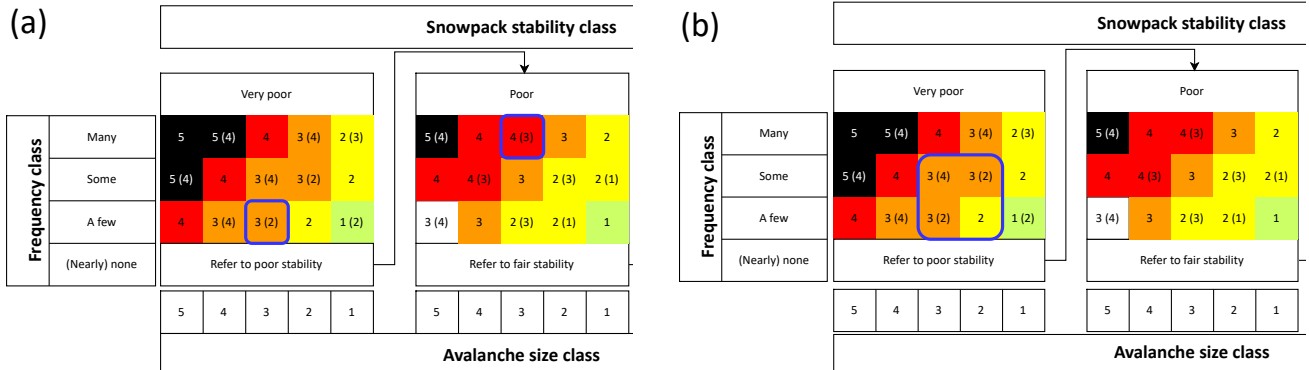

**Figure 9.** EAWS Matrix (extract) with several cells marked as relevant (blue border) to accommodate (a) a range of possible combinations or (b) uncertainty. In (a), two combinations of stability and frequency are considered (*very poor - a few*, *poor - some*), in (b) uncertainty relates to frequency (*a few* or *some*) and avalanche size (*size 2 or 3*). For explanations refer to text.

Another potential cause impacting a reliable estimation of the factors is linked to the workflow and design of the Matrix,
which require a forecaster to choose exactly one single cell in the matrix. However, there may be situations when multiple combinations describe the prevailing or anticipated avalanche conditions equally well, or when a forecaster is uncertain with regard to selecting a specific class. In both these situations, a forecaster has to settle on one option only, which is a much less flexible approach compared to the approach used in North America (Statham et al., 2018), where forecasters can mark several cells. A more informative approach would be to allow forecasters to mark all combinations in the matrix, which are considered
relevant for the given situation, rather than mandating a single choice. For example, imagine that *poor* stability is expected on *many* slopes with avalanches up to *size 3*. In addition, in *a few* places stability is considered *very poor*, as natural avalanches are expected (Fig. 9a). In this case, the forecaster would highlight these two cells. A similar approach could also be employed to convey uncertainty (Figure 9b). For example, a forecaster may be confident that stability is *very poor* but may be uncertain whether the frequency of these locations should be considered *a few* or *some*, and whether avalanche size will be size 2 or size
3. In such cases, the respective cells in the matrix could be marked, expressing the uncertainty of these classes. Ideally, more information relevant to assess these factors should be collected to reduce the uncertainty associated with these labels. In both cases, rules for selecting $D$ need to be established, such as selecting the cell resulting in the highest danger level in case of the first scenario (*poor-many - 3* with $D = 4(3)$, Fig. 9a), or by choosing the median or majority of the indicated $D$ in the second case ($D = 3$, Fig. 9b). In addition, communication with forecast users should convey this information in a clear manner, for
instance, by stating that both natural and human-triggered avalanches are possible in scenario 1.

By imposing a single discrete choice on forecasters, uncertainty inherent in the assessment process may inadvertently be masked. Moreover, forcing forecasters to express their detailed judgments by forcing them to choose from a small number of discrete choices may violate the basic maxim of forecasting postulated by Murphy (1993, p. 282), namely that a «forecast should always correspond to a forecasters best judgement». Assume, for example, that a forecaster expects a range of avalanche
conditions within the forecast domain, but that spatial variations cannot be expressed with this level of detail in the forecast.





This results in a mismatch between the forecasters best judgement and the forecast, requiring the forecaster to simplify and deviate from the best-possible estimate, inevitably impacting the accuracy of the forecast (e.g., Murphy (1993), Techel (2020, p. 72-74)). Thus, to reduce a potential mismatch between the forecasters best judgment and the forecast, an alternative way could be to express the tendency within a class or the uncertainty between choosing one of two options by either using a finer-grained

scale (more classes, more spatial units) or by allowing for sub-categories within existing classes. For instance, a forecaster might choose a class, such as *a few*, but indicate a tendency toward the next class, such as *some* by using *a few→some*. Using more classes competes with the human capability of reliably being able to choose at most from seven classes (e.g., Miller, 1956; Kahneman et al., 2021). Moreover, each class would need a clear definition. However, more promising is combining absolute and relative judgments (e.g., Kahneman et al., 2021), as this capitalizes on the fact that humans are generally good at

making relative rankings within classes established before-hand. Such concepts are already used in avalanche forecasting. For instance, in North America, it is common that avalanche practitioners assess avalanche size using intermediate classes. While there are definitions for avalanches of size 2 and size 3 (i.e. Table 4), a size 2.5 is simply in between these classes (e.g., Hafner et al., 2023). Similarly, in Switzerland, sub-levels indicate whether avalanche danger is expected low, in the middle, or high within a level, requiring the forecaster to first make an absolute judgment ($D$), and then a relative ranking (Techel et al., 2022;

Lucas et al., 2023).

To ensure a consistent and practical application of the EAWS Matrix, it is crucial that the definitions for the factors are clear and easily applicable in an operational setting. For instance, avalanche size is defined based on physical measurements, such as volume or mass, or by the destructive potential of the avalanche. Similarly, the definition of snowpack stability is closely linked to typical observations. Thus, both avalanche size and snowpack stability can be assessed using observational data. In

contrast, it is difficult to unambiguously define frequency classes, particularly when considering that the frequency of locations with *very poor* or *poor* stability is generally low. Given clear definitions, it would be imminent that all forecasters possess a sound understanding of and consistently adhere to these. However, even if definitions for all factors and their classes were clear, inconsistencies cannot be completely eliminated. The interpretation of current conditions, often based on limited observations, combined with the inherent uncertainties in numerical weather prediction models, will inevitably lead to slight variations in

interpretation among forecasters. Thus, inconsistency also becomes a function of data availability and reliability.

## 5.4   Towards an updated avalanche danger scale

The EAWS Matrix and related look-up tables have undergone continuous development over the past two decades. The European avalanche danger scale, in contrast, has remained unchanged since 1993. As a result, EADS and EAWS Matrix, initially closely linked, have gradually diverged. To ensure a unified and coherent framework for avalanche danger assessment and

communication, it is essential to reestablish a robust connection between them.

The survey on the EAWS Matrix and data on its operational usage provide valuable insights on how avalanche danger levels are typically described in forecast products. Drawing on this information, the connection between the EADS and the EAWS Matrix can be reestablished by updating the EADS using the terms typically chosen in the Matrix to describe the typical character of every avalanche danger level.





Figure 8 showed that there is a high level of agreement among warning services regarding the typical description of the danger levels using the factors. Considering the median value as a representative and consensus-oriented approximation to describe a factor or factor combination for a specific danger level, we summarized the warning service specific descriptions of avalanche danger (Fig. 8) in Table 8. For comparison, we also show the respective combinations for the EAWS Matrix and EADS. As can be seen, there are some differences in what is the typical class for each factor for each danger level across all

three sources. $D = 4$ shows the least variation. For $D = 1$ and $D = 2$ we see the previously mentioned tendency to emphasize *poor* stability, combined with either an avalanche *size 1* or the frequency class *a few*, instead of *fair* stability as in the EADS. $D = 3$ has a tendency to *size 2* avalanches in the usage data compared to the EADS and the Matrix. The differentiation between $D = 2$ and $D = 3$ is substantially less pronounced in the usage data compared to the Matrix and especially to the EADS.

If factors were used similarly by European forecasters and if the distribution of cases within a danger level were approxi-

mately balanced in the usage data, it would be possible to derive an updated danger scale based on usage patterns. However, as seen in data from Switzerland (Sect. 4.1), cases tended to be more frequent at the respective lower end of danger levels 3-considerable and 4-high, biasing the usage data towards the lower end of these danger levels. It is important to recognize that winter seasons vary in terms of snowfall, the frequency and duration of cold spells, and other meteorological factors. Consequently, these variations lead to differences in avalanche conditions, the severity of avalanche problems, and ultimately

the assessment of avalanche danger. Our usage data is based on a single winter season, which may have been influenced by the unique characteristics of that particular season. Thus, data from more seasons including information on sub-classes of the factors and sub-levels of $D$ need to be analyzed before an updated danger scale can be derived based on the usage data.

### 5.5 Recommendations for practice and ways forward

We identified areas for improvement in the current standards and definitions used in avalanche forecasting (Sect. 2) but also

with regard to the operational use of the Matrix. The latter relate primarily to the inconsistent assignment of factor classes but also to the design of the Matrix. From these, the following recommendations and potential areas for further development emerge with the goal to reduce the variations due to an individual forecaster or warning service preparing a forecast, and, thus, to ultimately lead to better forecast products for forecast users:

1. **Increase reliability of estimation of factors**: Particularly in warning services where forecasters tend to work primarily

by themselves, regular training sessions are essential to foster a common understanding of factor categories. This will permit calibration in the use of the categories between forecasters. Ideally, such exchanges would also occur across warning service boundaries, to develop and maintain a common understanding of categories.

2. **Evaluate methodologies to estimate factors**: It should be evaluated whether choosing multiple cells in the matrix (as discussed in Sect. 5.3 or as used in the hazard chart in North America, Statham et al., 2018), or whether continuous

sliders or sub-categories are alternative and more effective ways to factor estimation compared to choosing a single category from a small number of classes (concept currently used in the EAWS Matrix).



3. **Evaluate alternative Matrix layouts**: Investigate different matrix layouts, such as used in ADAM (Müller et al., 2016, see also Fig. A2) or the data-driven matrix by Techel et al. (2020a, see also Fig. A4), to determine if they offer advantages over the current design of the EAWS Matrix.

4. **Revise Matrix based on data**: The Matrix is based on expert estimates. However, data-driven analyses describing danger levels using a variety of relevant data sources should be leveraged, as for instance the comprehensive study by Techel et al. (2022), which provides insights into how observations and model predictions relate to forecast danger levels. Additionally, data sources such as avalanche detection data from satellites or terrestrial systems (e.g., Eckerstorfer et al., 2017; Mayer et al., 2020) or snowpack modelling combined with machine-learning approaches (e.g., Techel et al., 2022; 615    Mayer et al., 2023) may be suitable to obtain data-driven versions of the Matrix.

5. **Reduce the subjective component of avalanche forecasting**: Currently, the quality of avalanche forecasts relies strongly on the competence and experience of a human forecaster interpreting a variety of heterogeneous data. However, with the increasing availability of reliable, highly-resolved models in combination with state-of-the art data analysis techniques (e.g., Herla et al., 2022), forecasting should become more data- and model-driven. This may be achieved by combining 620    model predictions and the forecaster's best judgment using hybrid intelligence methods (Dellermann et al., 2019). Moreover, using historic data and machine-learning approaches, forecaster decisions may be directly modeled (Pérez-Guillén et al., 2022; Maissen et al., 2024), providing the forecaster with a data-driven second opinion when choosing a danger level.

6. **Validate the «correctness» of forecast factor categories and danger levels**: Although difficult to achieve due to a 625    general lack of relevant data, the evaluation of the performance of avalanche forecasts should become standard practice. Such evaluations should go beyond the evaluation of the «correctness» of forecast danger levels (as done in several previous studies, e.g., Techel and Schweizer, 2017; Logan and Greene, 2023), but should consider all published forecast parameters as emphasized by Lucas et al. (2023). As validation data is scarce, data sources like automated avalanche detection data or snowpack modelling should be explored. Such evaluations should be ongoing to maintain forecast 630    quality, but, importantly, these should be made before introducing new parameters in avalanche forecasts.

7. **Update avalanche danger scale**: Clearly, congruence between definitions, workflow, EAWS Matrix and the EADS is of utmost importance. Therefore, the EADS should be updated using available data, including usage data as described in Section 4.1.




**Table 8.** Characterization of danger levels as described in the EADS and EAWS Matrix, and the most frequent combinations used during the season 2022/2023. The latter summarizes the results shown in Figure 8 for the seven groups of warning services, with the most frequent factor shown in the column *individual* and the most frequent combination of factors in the column *combined*. The first value indicates the most frequently used class or combination, values in brackets indicate if a second class or combination was associated with a danger level more than 30% of the time.

| D | Matrix use - combined | | | Matrix use - individual | | | EAWS Matrix | | | EADS | | |
|---|---|---|---|---|---|---|---|---|---|---|---|---|
| | stab | freq | size | stab | freq | size | stab | freq | size | stab | freq | size |
| 1 (low) | F (or P) | Fe | 1 | F (or P) | Fe | 1 | F | Fe | 1 | F | Fe | 1 or 2 |
| 2 (moderate) | P | So (or Fe) | 2 | P | Fe (or So) | 2 | P or F | So (or Fe) | 2 | F | So | n.d., <4 |
| 3 (considerable) | P | So | 3 | P (or VP) | So | 2 (or 3) | P | So | 3 | P | So or Ma | ≤ 3 or 4 |
| 4 (high) | VP | Ma | 3 | VP | Ma | 3 | VP | Ma | 3 or 4 | VP (or P) | Ma | 3 or 4 |
| 5 (very high) | – | – | - | – | – | - | VP | Ma | 5 | VP | Ma | 4 or 5 |

Stability (stab): fair (F), poor (P), very poor (VP); frequency (freq): a few (Fe), some (So), many (Ma); avalanche size (size): 1 - 5.





# 6 Conclusions

Avalanche forecasting involves both the assessment of the current and future avalanche conditions as well as their communication to the public. Avalanche forecasting is currently of categorical nature (e.g. factors, avalanche problems and danger level). The quality of a forecast depends on data availability, the skill of the forecaster, and the definition and understanding of the categories used. In this study we investigated the latter by evaluating the updated EAWS Matrix and its associated factors and ascertain the extent of consistency in their understanding and application.

The updated EAWS Matrix, as presented in Section 4, represents the latest consensus among European avalanche forecasters. The EAWS Matrix has been favorably received as a valuable tool in operational avalanche forecasting and in forecaster training. It offers a standardized framework for evaluating avalanche danger by linking three key factors: snowpack stability, frequency of snowpack stability, and avalanche size, to the danger level ($D$). The introduction of a second danger level ($D^2$) acknowledges the remaining inconsistencies among avalanche forecasters and combinations with limited response rates ($\leq 70\%$) in the matrix

survey signify rare or unconventional scenarios, as perceived by most forecasters (Figure 3).

Our analysis of the EAWS Matrix's implementation by 15 European avalanche warning services across eight countries during its inaugural winter unveiled a generally consistent pattern in class assignments to each danger level. However, consistency among forecasters when choosing individual factors is currently relatively low, necessitating refinements in definitions and the workflow for avalanche danger assessment. In addition, training sessions are essential to improve consistency and enhance

forecaster skills in applying the EAWS Matrix effectively. Diverging trends between services currently still exist, leading to an overlap of factors used at different danger levels. Notably, distinguishing between danger levels 2 and 3 posed more challenges compared to levels 1 and 2, and 3 and 4 (no data was available for level 5). While a revision of the EAWS Matrix after a single winter is challenging due to seasonal variations, our goal is to find a consensus that suggests one danger level per cell to reduce inconsistency to a minimum in the future.

When applying the EAWS Matrix, we recommend to assign multiple cells where applicable for describing avalanche conditions to account for complexity and uncertainty and avert from the practise of enforcing the assignment of a single combination. Further, considering categories on a more nuanced scale can contribute to a more consistent assessment of avalanche danger level, provided that classes can be clearly defined and assessed. Especially a better understanding and more accurate measurement of frequency of snowpack stability are crucial for improved consistency.

Currently, the updated definitions, workflow, and EAWS Matrix presented by Müller et al. (2023) represent a departure from the European avalanche danger scale. Using repeatedly collected data on matrix usage, along with ongoing discussions and training among European avalanche forecasters, can provide a robust foundation for developing an updated avalanche danger scale incorporating the defined terms.



*Code and data availability.* Data and code will be published at the repository envidat.org, and will also be indexed at https://opendata.swiss/
en/.



## Appendix A: Avalanche Danger Matrices



**Figure A1.** The figure depicts the EAWS-Matrix-v2017 (figure taken from EAWS, 2017), which is essentially identical to the Bavarian Matrix (BM EAWS, 2005) when considering the large cells only. The BM illustrates the distribution of hazardous zones against the probability of avalanche release. Initially, the left-hand side, which dealt with artificially triggered avalanches, lacked the factor of avalanche size entirely. Meanwhile, the right-hand side focused on naturally triggered avalanches, providing indications of expected sizes in the column headings. However, an update in 2017 integrated avalanche size (small cells) into the left-hand side as a third dimension, leading to its transformation into the EAWS-Matrix-v2017 (EAWS, 2017).

**Figure A2.** The Avalanche Danger Assessment Matrix (ADAM), as published by Müller et al. (2016), presents two versions: one aligning with the terminology of the European Avalanche Danger Scale (EADS, Table 1) at the top and another adhering to the Conceptual Model of Avalanche Hazard (CMAH) at the bottom. ADAM consists of a likelihood matrix (left-hand side), which defines likelihood terms based on the spatial distribution and snowpack stability (sensitivity to trigger), and the Danger Matrix (right-hand side), which provides guidelines for determining the appropriate danger level by combining the likelihood of triggering with avalanche size.




| | Size D1 | Size D2 | Size D3 | Size D4 | Size D5 |
|---|---|---|---|---|---|
| **Strong (> 30%)** | 1 or 2 | 3 | 4 | 5 | 5 |
| **Good (10 to 30%)** | 1 | 2 or 3 | 3 or 4 | 4 | 5 |
| **Fair (1 to 10%)** | 1 | 1 or 2 | 3 | 4 | 4 |
| **Small (1 to 3%)** | 1 | 1 | 2 | 3 | 4 |
| **Slight (< 1%)** | 1 | 1 | 1 | 2 | 3 |

FIG. 2: GUIDANCE FOR COMBINING LIKELIHOOD OF AVALANCHES WITH AVALANCHE SIZE TO ASSIGN AVALANCHE HAZARD RATINGS (AFTER MULLER ET AL., 2016A; CLARK AND HAEGELI, 2018).

**Figure A3.** The likelihood matrix proposed by Thumlert et al. (2020) addresses concerns highlighted in their survey, revealing a broad spectrum of interpretations of likelihood terms outlined in the Conceptual Model of Avalanche Hazard (CMAH, Statham et al. (2018)) among avalanche professionals. The aim was to move away from likelihood terms commonly linked with higher probabilities and instead introduce levels of chance paired with a percentage range. The resulting likelihood matrix shown here bears resemblance to the danger matrix on the right-hand side of ADAM Müller et al. (2016). Intermediate levels are recommended for situations with higher chances and smaller avalanches.





**a)**
***stability matrix***

| snowpack stability | | frequency | | | |
|---|---|---|---|---|---|
| | | none* | few | several | many |
| | very poor | ** | D | B | A |
| | poor | ** | E | D | C |
| | fair | - | - | E | E |
| | good | - | - | - | - |

\* none or nearly none

\*\* if none, refer to next higher stability class

- no data

C cell contains less than 1% of the data

**b)**
***danger matrix***

| stability matrix | largest avalanche size | | | |
|---|---|---|---|---|
| | 1 | 2 | 3 | 4 |
| A | 3 -4 | 4 (-3) | 4 | 4 |
| B | 3 (-2/-1) | 3 (-2) | 3 (-2) | 4 -3 |
| C | 2 (-3) | 2 -3 | 3 -2 | - |
| D | 1 -2 | 2 -1 | 2 -1 | 3 (-2) |
| E | 1 | 1 (-2) | 1 (-2) | - |

-3:  >30%

(-3):  15-30%

**Figure A4.** Data-driven look-up table for avalanche danger assessment (figure and caption taken from Techel et al., 2020a). The (a) *stability matrix* combines the frequency class of the most unfavorable snowpack stability class (columns) and the snowpack stability class (rows) to obtain a letter describing specific stability situations, the (b) *danger matrix* combines the largest avalanche size (columns) and the specific stability situations (letter) obtained in the stability matrix (rows) to assess the danger level. In (b): The most frequent danger level is shown in bold. If the second most frequent danger level was present more than 30% of the cases, the value is shown with no brackets, if present between 15 and 30% it is placed in brackets. In (a) and (b): Cells containing less than 1% of the data are marked.



*Author contributions.* KM (project lead, study design, data curation, writing, reviewing), FT (study design, data curation, formal analysis, writing, reviewing), CM (study design, data curation, writing, reviewing).

*Competing interests.* We declare that they have no conflict of interest.

*Acknowledgements.* We thank Stefano Sofia, Petter Palmgren, Anne Dufour, Nicolas Roux, Fabiano Monti, Giacomo Villa, Guilem Martin Bellido, and Lorenzo Bertranda for invaluable discussions within the EAWS working group *Matrix & Scale*. We thank the EAWS workgroup coordinators Thomas Stucki and Thomas Feistl for their support over the years we were working on the EAWS Matrix.



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
