# Peer review of "The EAWS matrix, a look-up table to determine the regional avalanche danger level (Part A): Conceptual development"

_Natural Hazards and Earth System Sciences, 2024_

## Referee Comment (RC1)

Review of Müller, K., Techel, F., and Mitterer, C.: An updated EAWS matrix to determine the avalanche danger level: derivation, usage, and consistency, Nat. Hazards Earth Syst. Sci. Discuss. [preprint], https://doi.org/10.5194/nhess-2024-48, in review, 2024.

By Grant Statham

**General comments**

This is an important paper that describes in detail the process used by the European Avalanche Warning Services (EAWS) to assess an avalanche danger rating, with the objective of adding more objective support to the assessment in order to improve consistency between forecasters and forecast regions. For years the Bavarian Matrix, then the EAWS Matrix were background documents used in Europe and hard to follow and understand. With this manuscript, it is now possible to closely study the EAWS method and thus the topic is relevant, an important contribution to the field and appropriate for publication in NHESS.

The method described is specifically for determining an avalanche danger rating using the European Avalanche Danger Scale (EADS), an ordinal scale of 1-5. Determining a danger rating is one specific niche of avalanche forecasting (public avalanche forecasting), and does not apply to the broader scope of avalanche forecasting which includes avalanche forecasting for ski areas, transportation, infrastructure, guided recreation, etc. It is important to recognize this specific focus of the EAWS matrix and that this method will not easily apply to other types of avalanche forecasting as it is independent of terrain, and terrain is fundamental to avalanche forecasting. This EAWS Matrix assesses avalanche danger ratings, not avalanches themselves.

The EAWS Matrix is intended to bring consistency to the process, which is necessary as currently there is inconsistency in the application of danger ratings between forecasters and forecast regions. Avalanche forecasting is done from observational evidence and has little to no quantitative support, thus the process is mostly judgement based. This paper implements the quantitative concept of "frequency of snow stability" which has been evolving over the past decade and is an excellent step forward that advances the challenge of assessing the probability of avalanche release.

As a general comment, I find the EAWS Matrix method a bit awkward and not intuitive to follow in practice. It seems that the method of frequency of snow stability is well suited to a future state of using snowpack models in order to model avalanche danger ratings – which is an excellent and necessary step. But in terms of using this method in day-to-day avalanche forecasting and applying observational evidence it seems the flow is awkward as shown in the manuscript. I think this is because the focus is on determining a danger rating between 1-5, rather than understanding the distribution of avalanche hazard across the landscape. It certainly works for improving consistency by having all users follow the same process to land on the same danger rating, and this is important and necessary.

**Specific comments**

**Scale** I think a discussion on scale and scale issues is necessary for this manuscript. The concept of scale, and in particular the "point scale" is missing, yet measurements at the point scale are fundamental to this proposed EAWS matrix. The proposed "frequency of snowpack stability" method is based on the frequency of point scale measurements, yet this is unclear because snow stability is traditionally thought

of as a "slope stability" estimate, and well-established systems exist in different countries for estimating it (CAA, 2016; AAA, 2022).

Reuter et al. (2015) state that *"snow slope stability describes the mechanical state of the snow cover on an inclined slope"* then further add that *"the link between point observations of snow stability and snow slope stability is not clear"* and that *"the point stability scale is not even well defined".* Reuter and Schweizer (2018) state *"a description of snow instability at the scale of the snow cover or a point in the terrain is much needed, yet presently lacking"* and present a first framework for doing so. The Conceptual Model of Avalanche Hazard (Statham et al., 2018) describes spatial scales used in avalanche forecasting, but the point scale is missing at the lowest end. Techel et al. (2020) state *"the probability of avalanche release refers to a specific location and relates to the local (or point) snow instability".* Thus, according to this, the probability of avalanche release can only be determined in a single spot, until these spots are combined using the frequency method proposed here. Schweizer et al. (2023) suggest a definition (redefinition) of snow stability as *"point snow stability refers to snowpack layering, propensity for failure initiation and onset of crack propagation"* and distinguishes this from "slope stability".

All of this to say that although the concept of snow stability as a point scale measurement has been in development for close to a decade, this is not well established in practice. The concept is crucial and needs to be included here as it is the basis for this method. Scale issues are fundamental to avalanche forecasting and thus some explanation is necessary in this manuscript and I suggest a section dedicated to scale issues with an explanation and definition of snow stability as a point scale measurement.

**Frequency of snowpack stability** This concept is awkward, and the method is mismatched with actual avalanche forecasting as is done today. This is in-part due to the issues described above, where snow stability is not well defined as a point scale observation and without that, the concept of frequency of snow stability is hard to grasp.

Frequency of snowpack stability is a quantitative method using the "frequency distribution of points of snowpack stability" as a statistical measure of probability [Line 185: $f(i) = n_i/n$]. This method assumes one knows both $n_i$ and $n$, both of which are impossible to know without snowpack modelling that is underlined across the landscape. This method is well suited to compute a danger rating but requires snowpack modelling to supply the necessary data and seems distant from actual avalanche forecasting as practiced currently, which uses observational data and requires answering the question of "where" along with developing a mental image of how snow stability is distributed across the terrain.

"Distribution" is a much better term for this concept in the English language and for avalanche forecasting specifically, where the question of how snow stability is distributed across the landscape is one of the most fundamental questions to be solved. Frequency of snowpack stability tries to partially solve this statistically, but the EAWS Matrix is being applied by forecasters using observational data.

The $f$ formula is sound, but it will better resonate with avalanche forecasters if the concept is termed "Distribution of Snowpack Stability". I suggest lines 181 – 186 be revised in such a way:

1. Snowpack stability (point scale)
2. Distribution of snowpack stability (estimated using Table 3, or calculated using frequency)
3. Avalanche size

This is consistent with using observational data, which is incompatible with the frequency formula. When the data is available, distribution can be calculated as a frequency using $f(i) = n_i/n$.

At the very least, using the full term "frequency distribution" will help as is how the concept is termed in Techel et al., 2020.

Lines 616-623 describe a future state where subjectivity is reduced in avalanche forecasting through the increased use of highly resolved models, which is an excellent objective but still under development. When the use of snowpack modelling becomes implemented operationally, then then $f(i) = n_i/n$ can be properly utilized.

**Lines 27-29** describe the preparation of regional avalanche forecasts as involving two steps: 1) assessment of current and future avalanche danger, and 2) communication of future avalanche danger. This is true for the assessment of avalanche danger, but not for the preparation of a regional avalanche forecast, which also includes (depending on region) analysis of snowpack, terrain, locations and avalanche problems. Avalanche danger is only one component of a regional avalanche forecast and this paper focusses on the assessment of avalanche danger ratings. Please revise this section to reflect a broader view of what preparing a regional avalanche forecast entails or narrow the scope of the preparation down to avalanche danger ratings only. This same issue is found on Line 67, where *"regional avalanche forecasting"* should be narrowed in scope to "the assessment of avalanche danger ratings" and Line 72 where *"the process of regional avalanche forecasting"* should also be narrowed in scope to "the standards for assessing avalanche danger ratings".

**Lines 47-49** describe the importance of reliable avalanche forecasts in reducing damage and loss, enhancing safety and mitigating risks. This is all correct but focuses solely on negative consequences and ignores positive outcomes, which is half of the risk equation and a major reason for public avalanche forecasts. It is not all about loss. In addition to preventing loss, avalanche forecasts enable backcountry experiences by highlighting good conditions and appropriate terrain choices, improving the experience for backcountry users. It is essential to not only on focus on mitigation and loss, but also on gain and enabling the backcountry experience. I would like to see this theme applied throughout this paper where appropriate.

**Line 117** a better heading would be "Factors and workflow to determine avalanche danger levels"

**Tables 2,3 & 4** I realize these are EAWS standards and perhaps not easily changed, but I suggest you consider the following to make these crucial reference tables more applicable in practice. These tables should provide enough reference to make assessments based on observational evidence.

Table 2 - In general I find this table insufficient for making a proper assessment of snow stability. EAWS 2022b Appendix A has much more detail and I suggest improving this paper's Table 2 by adding additional columns to aid in assessment (i.e., stability test results).
— Make it clear that this is a point scale assessment.
— Change the Description of Very Poor to "very easy to trigger (e.g., natural)" to be consistent

Table 3 – Consider showing the frequency equation $f(i) = n_i/n$ *and* add a column on the right showing typical $n_i/n$ values

Table 4 – Rephrase Size 1 be more concise, "Unlikely to bury or injure a person except in terrain traps".

**Line 430** – The SWI forecasters were instructed to assess the three contributing factors, but to then ignore the EAWS Matrix and this led to a 46% deviation from the EAWS Matrix. This is a substantial deviation and makes me wonder why. Please explain why there is such a large deviation. Also Line 433 says that "forecasters largely concurred with the danger level proposed by the EAWS Matrix", except for SWI, which is the only service not linked to the matrix, where they disagreed 46% of the time. I think this needs more explanation please.

**Line 443** – This is just a comment, but the fact that "All warning services described stability most often as *poor* when giving D=2" in interesting because typically at D2, avalanche conditions are generally stable except for a few locations, thus backcountry travel can be done in many locations. This scenario shows that forecasters are assessing this D2 based on the frequency of *poor* stability. Although this Matrix applies the concept of snow stability, the assessments of avalanche danger ratings are actually determined and driven by the frequency of *instability*. This links to my earlier comments about Lines 47-49 whereby this model is driven by negative consequences, which is a bias towards instability and fails to appreciate stable conditions. No change requested, just an observation which has followed me through this review and has made me wonder why stability is not instead referred to as instability, when this is what truly drives the D assessment?

**Technical corrections**

**Line 6** – replace *"ensure"* with "promote" or "improve" as these look-up tables will not ensure or guarantee consistency.

**Line 19** – rephrase "*further efforts are required to develop and implement regional avalanche forecasting standards to reach the goal of forecasts being a reliable, credible, and timely source of information of expected avalanche conditions…"*. This reads like the goal of being reliable, credible and timely has not yet been achieved, which is incorrect. Consider "*further efforts are required to develop and implement regional avalanche forecasting standards to improve the reliability, credibility and timeliness of avalanche forecasts, regardless of forecaster or warning service behind the product".*

**Line 44** – "*forecasters"* should be possessive (i.e.: forecaster's).

**Line 88** – replace *"Despite"* with "Except".

**Line 108** – spatial distribution should be "frequency"

**Line 131** – Replace reference to Statham et al., 2010 with: https://arc.lib.montana.edu/snow-science/objects/issw-1996-060-066.pdf

**Line 158** – typo "allowed forecasters to adjust…"

**Line 220** – delete "In case that avalanches release" and just say "Avalanches can reach up to *size 3*.

**Line 279** – add "ratings" to the references to avalanche danger. This matrix determines avalanche danger ratings, not avalanche danger.

**Line 298** – This workflow description appears to be independent of terrain. Please indicated where is terrain (location) is assessed and how does this fit into the workflow?

**Figure 3** – This is just a comment and not a request to change anything, but I did find the matrix easier to use when I inserted the avalanche activity examples from the stability classes shown in Table 2. The figure below shows what I mean (red text). When I make an assessment, and consider the stability class as a point, then having my estimate of what could trigger an avalanche at that point is helpful and easy to cross reference with the terms many, some, a few, none. Simple, but it makes the assessment more intuitive and easier for me.

[Figure]

**Figure 3 caption** – "first assess the frequency of the most vulnerable locations". The most vulnerable locations is not consistent Line 285 that says "they begin by considering the lowest stability class". What do you mean by "the most vulnerable locations"? This is even harder to resolve when stability is defined as a point. This is also the only reference I have seen with regards to locations (i.e., terrain) and this model seems independent of terrain.

**Line 306** – rephrase " and when spatially continuous" to better explain what this means.

**Line 561** – change the word "imminent". Do you mean "important"?

**Line 635** – change the start of the sentence to "Public avalanche forecasting …" as some kinds of avalanche forecasting do not involve public communication.

**Conclusion**

Thanks for the opportunity to review this manuscript. I think it's an excellent development and advances some of the concepts that the public avalanche forecasting community have been struggling with. Most importantly I am encouraged by the implementation of snow stability as a point measurement, which leads to the concept of frequency distribution as a method to determine avalanche release probability. Probability of avalanche release is a vexing problem to solve, and this work positions the concept well for a future of determining avalanche danger ratings using models. This is an important development.

Sincerely,

Grant Statham
June 24, 2024

---

## Referee Comment (RC2)

Review of *An updated EAWS matrix to determine the avalanche danger level: derivation, usage, and consistency* by Müller et al. (nhess-2024-48)

**General Comments**

In this study, the authors provided a thorough background on the European Avalanche Danger Scale (EADS) and European Avalanche Warning Services (EAWS) Matrix development and definitions, described the revised EAWS Matrix and associated methods of updating the Matrix, and then presented relevant results of using the newly revised EAWS Matrix for one full season. This manuscript is well written and logically organized. The methods are sound, and the results are supported by sufficient evidence. The interpretation of those results is reasonable with reference to existing literature. The recommendations section of this paper is very useful and points to limitations of consistency by providing solutions. Overall, I think this is a valuable contribution to the literature, fit well within this Special Issue, and should be published. I have a couple of general comments for the authors to consider and a few specific and technical comments as well.

I understand that this study focuses on evaluating the updated (current) EAWS matrix, its use and consistency among forecasters, and compatibility with EADS, and not necessarily an evaluation of the individual three key factors that determine the danger rating. As the authors point out, consistency in forecasters' evaluation or interpretation of the three key factors is crucial for the matrix to be used to its full potential. Indeed, the authors provide recommendations on how to enhance the use of the matrix by improving consistency in the three contributing factors. However, it seems that across the surveyed regions, a variety of input data (in-situ observations, model output, meteorological data, etc.) exist. The authors mention the influence of input data very briefly (lines 485-489), but the quality and quantity of input data plays a crucial role in danger assessment and would potentially influence the assessment variability across forecasters and regions. Can the authors provide information or comment on, generally, what data types each region uses and/or provide evidence on how the assimilation process of various types of data across regions may influence the classification of the three factors and ultimately the danger rating?

The authors present a thorough summary on the evolution of EAWS and the EADS and provide some geographic references to other non-European forecasting tools like the CMAH in North America. However, there are no references to other avalanche sector decision making tools. In other words, this study focuses on public avalanche forecasting operations, but not forecasting in other sectors like transportation corridors, ski areas, natural resource industry, etc. Is the same tool used for those sectors throughout Europe? I suggest being clear that this study focuses on a matrix for public avalanche forecasting or state how using the EAWS matrix in those sectors differs, if at all, from public avalanche forecasting.

**Specific and Technical Comments**

Figure 2: Consider adding a legend to the proportion scales for b) and c) that easily shows the reader which colors represent higher correlation values.

Figure 6: The use of $D^1$ and $D^2$ here is confusing. Is this the same as the median $D^1$ and $D^2$ used in Figure 2a and defined in Lines 262 -264? Also, if there was disagreement with D (forecaster derived) and $D^1$ (Matrix derived), that is indicated in the left column in Figure 6, correct? If the second column represents $D^1 \neq D^2$, then forecasters used $D^2$ (again the median second selection from Figure 2a)? Please clarify.

Line 88: four? In "Despite for minor changes in 1994…" or do you mean "Except for minor changes…"

Line 104: the way this is written is confusing to me. I read it as 'as stability decreases you need a greater load to trigger an avalanche.' (i.e. inverse relationship). Perhaps 'instability' should be used here instead of stability.

Lines 241 and 244: 60 responses in total in line 241, but in line 244, you state 76 responses. What is the difference?

Line 258: Similar to the comment above (Line 104). I view an increase in stability as the snowpack becoming more stable. Instability?

Line 345: "center of gravity"? Do you mean largest proportion?

---

## Author Response (AR1)

**Reply to editor**

Dear Pascal,

We have revised our manuscript in response to the thorough reviews provided by Grant Statham and Erich Peitzsch. We applied the technical and specific corrections suggested by the reviewers as stated in our earlier author responses (attached below). Additionally, we reviewed our manuscript and adjusted language and grammar where we saw fit (without changing meaning or context). For a detailed comparison, please refer to the attached latexdiff.

Below, we outline the more substantial revisions we made in alignment with the general comments from both reviewers. Line numbers refer to the updated manuscript for clarity.

We sincerely appreciate the valuable input and patience from you and the reviewers throughout this process. We hope the revised manuscript satisfactorily addresses all comments.

Best regards and a Happy New Year,

Karsten, Frank, and Christoph

**General comment**

Erich Peitzsch commented that the focus of our study is regional avalanche forecasting and the consistent determination of the avalanche danger level. We use now "avalanche danger level" explicitly through the text wherever applicable e.g.,

**Line 57:** "With the aim to increase consistency between forecasters and warning services when deciding on an avalanche danger level for a region,..."

Line 116: Section title "Factors and workflow to determine avalanche danger levels"
* * *
We addressed the general comment on "scale" by Grant Statham in the following sections:

Line 181-191: "In theory, the EAWS workflow requires forecasters to estimate the frequency distribution of snowpack stability classes across all points in avalanche terrain within a warning region. Independent of the spatial scale of the forecasting problem, assessing snowpack stability has traditionally relied heavily on observations of avalanche activity, signs of instability, and stability test results (Reuter and Schweizer, 2018). More recently, this has been complemented by stability information extracted from one-dimensional physical snowpack models (e.g., Mayer et al., 2022). In practice, estimating snowpack stability at every point in a large region is yet impossible. Forecasters therefore infer the distribution of stability classes across a region by combining sparse point observations and model data (when available), and their expertise and intuition. The estimated proportion of potentially unstable points, relative to a specific triggering level, reflects the likelihood of triggering an avalanche at a random point in avalanche terrain. This likelihood, combined with the potential avalanche size, determines the regional danger level. This approach aligns with the hazard chart in the CMAH, which categorizes avalanche danger based on the likelihood and size of avalanches (Statham et al., 2018)."

We added another bullet point to our list of recommendations in the discussion.

**Lines 631-633:** "Adjust size of warning regions: Spatial units for regional forecasting should be of comparable size and fall within a standardized range of square kilometers, ensuring consistency. Their boundaries should be determined based on the availability and density of relevant data."
* * *
We addressed the general comment on "frequency distribution" by Grant Statham by consistently using the term "frequency distribution" throughout the text. We also updated and extended the "Background" section to highlight the difference between frequency distribution of snowpack stability and spatial distribution of avalanche problems:

Lines 192ff: "Avalanche problems, such as persistent weak layers or wind slabs, describe typical avalanche scenarios and are integral to avalanche danger assessment. However, these problems do not directly correspond to specific snowpack stability classes, which vary spatially and temporally. For example, a persistent weak layer may be widespread, but snowpack stability could range from very poor in a few locations to poor or even fair elsewhere. Thus, the presence of an avalanche problem does not necessarily equate to a specific snowpack stability class at any given location. While the EAWS Matrix focuses on the frequency distribution of snowpack stability classes associated with specific avalanche problems, the CMAH emphasizes the spatial distribution of avalanche problems. This distinction highlights the key difference between these two approaches currently in use."

**Lines 317-320:** "The workflow is specifically designed for regional avalanche forecasters. It assumes that the forecast area is large enough to encompass multiple mountains, elevation zones, all aspects, and varied terrain features, such as ridges, gullies, and open slopes. Consequently, terrain is not treated as an independent factor."
* * *
We added a separate section on "Decomposing the avalanche forecasting task".

**Lines 220-235:** "The workflow introduced by the CMAH and adopted by EAWS decomposes regional avalanche forecasting into smaller, more manageable components. This decomposition is generally expected to improve the accuracy of forecasts, as breaking down complex tasks can lead to more precise estimates (MacGregor, 2001). However, as noted by MacGregor (2001, p. 107), "Decomposition should be used only when the estimator can make component estimates more accurately or more confidently than the target estimate.

The accuracy of human estimates depends on various factors, including the quality of available data, the forecaster's expertise in interpreting the data, and their understanding and consistent application of the predefined categories (Stewart, 2001). In a regional avalanche forecasting context, these categories describe snowpack stability, the frequency of the lowest stability class, and the largest expected avalanche size (Tables 2-4), the key inputs to the EAWS Matrix. Inconsistencies in category assignments among forecasters can reduce the quality of the resulting forecast, particularly the accuracy of the avalanche danger level (Murphy, 1993; Techel et al., 2024)."
* * *
We tried to explain the deviations seen in the Swiss data in more detail as requested by Grant Statham:

**Lines 444-447:** "In Switzerland, however, forecasters assessed factors and D independently, without directly referencing the EAWS Matrix. This approach is similar to the North American

system, where factors (e.g., likelihood and avalanche size) are evaluated for each avalanche problem on the hazard chart, but where the final decision on D remains with the forecaster as no clear assessment rules are defined (Clark and Haegeli, 2018). Comparing Swiss forecasters choice of D given the selected factor combination, D frequently deviated from D1 in the EAWS Matrix. However, absolute differences were generally minor as forecasters most frequently selected the sub-level closest to D1 (Techel et al., 2024, for a description of sub-levels used in Switzerland refer to Techel et al. (2022))."
* * *
Figure 2: Capital letter in axis title: "Avalanche size".

Figure 6: We changed the column titles to "Adhere to D1" and "Deviate from D1".

We re-named figures according to their appearance in the text by adding "fig##\_" as a prefix to the filename.
* * *
PS: We attached our earlier replies to the two reviewers for completeness on the following pages.

**Reply to Grant Statham**

We sincerely thank Grant Statham for his thorough and constructive review. Please find our answers to each of the points raised below.

**General comments**

**Scale**

I think a discussion on scale and scale issues is necessary for this manuscript. The concept of scale, and in particular the "point scale" is missing, yet measurements at the point scale are fundamental to this proposed EAWS matrix. The proposed "frequency of snowpack stability" method is based on the frequency of point scale measurements, yet this is unclear because snow stability is traditionally thought of as a "slope stability" estimate, and well-established systems exist in different countries for estimating it (CAA, 2016; AAA, 2022).

Reuter et al. (2015) state that "snow slope stability describes the mechanical state of the snow cover on an inclined slope" then further add that "the link between point observations of snow stability and snow slope stability is not clear" and that "the point stability scale is not even well defined". Reuter and Schweizer (2018) state "a description of snow instability at the scale of the snow cover or a point in the terrain is much needed, yet presently lacking" and present a first framework for doing so. The Conceptual Model of Avalanche Hazard (Statham et al., 2018) describes spatial scales used in avalanche forecasting, but the point scale is missing at the lowest end. Techel et al. (2020) state "the probability of avalanche release refers to a specific location and relates to the local (or point) snow instability". Thus, according to this, the probability of avalanche release can only be determined in a single spot, until these spots are combined using the frequency method proposed here. Schweizer et al. (2023) suggest a definition (redefinition) of snow stability as "point snow stability refers to snowpack layering, propensity for failure initiation and onset of crack propagation" and distinguishes this from "slope stability". All of this to say that although the concept of snow stability as a point scale measurement has been in development for close to a decade, this is not well established in practice. The concept is crucial and needs to be included here as it is the basis for this method. Scale issues are fundamental to avalanche forecasting and thus some explanation is necessary in this manuscript and I suggest a section dedicated to scale issues with an explanation and definition of snow stability as a point scale measurement.

I think a discussion on scale and scale issues is necessary for this manuscript. The concept of scale, and in particular the "point scale" is missing, yet measurements at the point scale are fundamental to this proposed EAWS matrix. The proposed "frequency of snowpack stability" method is based on the frequency of point scale measurements, yet this is unclear because snow stability is traditionally thought of as a "slope stability" estimate, and well-established systems exist in different countries for estimating it (CAA, 2016; AAA, 2022).

We agree that a standard definition of the point scale is currently lacking. However, we have the impression that *point scale* is the commonly understood term when referring to the evaluation of snowpack stability. We will provide or reference a definition of the *point scale* and incorporate it into the concept of *frequency [distribution]* of snowpack stability. Avalanche danger can be

assessed across different scales. Regional public avalanche warnings are typically provided for areas of 100-300 km2 and larger. Site-specific warnings are generally issued for one or several avalanche paths, e.g., along a road section. Snowpack stability, however, is assessed at a single point - usually by digging a snow pit to record a snow profile or conducting stability tests (e.g. ECT, Rutschblock). The extent of this assessment is typically no more than about 2x2 m, which we define as the point scale. Several studies validating avalanche danger levels rely on the frequency distribution of snowpack stability at the point scale (e.g., Schweizer et al., 2003, Bakermans et al., 2010). In theory, the EAWS workflow requires estimating stability assessments at all points within a warning region. A higher frequency of unstable points corresponds to a higher likelihood of triggering avalanches. This frequency distribution of stability classes (representing likelihood), combined with the potential avalanche size, determines the danger level for the region — a concept similar to the hazard chart in the CMAH. In practice it is impossible to assess the snowpack stability at all points within a region of several 100 km2. Forecasters must extrapolate and imagine the frequency distribution of unstable points based on available observations and models. While the CMAH uses the spatial distribution of avalanche problems to assess danger, an avalanche problem does not necessarily correspond to a fixed stability class in space or time. Therefore, the presence of an avalanche problem is not directly equivalent to the snowpack stability at a specific location.

**Frequency of snowpack stability**

This concept is awkward, and the method is mismatched with actual avalanche forecasting as is done today. This is in-part due to the issues described above, where snow stability is not well defined as a point scale observation and without that, the concept of frequency of snow stability is hard to grasp. Frequency of snowpack stability is a quantitative method using the "frequency distribution of points of snowpack stability" as a statistical measure of probability [Line 185: f (i) = ni /n]. This method assumes one knows both ni and n, both of which are impossible to know without snowpack modelling that is distributed across the landscape. This method is well suited to compute a danger rating but requires snowpack modelling to supply the necessary data and seems distant from actual avalanche forecasting as practiced currently, which uses observational data and requires answering the question of "where" along with developing a mental image of how snow stability is distributed across the terrain. "Distribution" is a much better term for this concept in the English language and for avalanche forecasting specifically, where the question of how snow stability is distributed across the landscape is one of the most fundamental questions to be solved. Frequency of snowpack stability tries to partially solve this statistically, but the EAWS Matrix is being applied by forecasters using observational data. The f formula is sound, but it will better resonate with avalanche forecasters if the concept is termed "Distribution of Snowpack Stability". I suggest lines 181 – 186 be revised in such a way:

- 1. Snowpack stability (point scale)
- 2. Distribution of snowpack stability (estimated using Table 3, or calculated using frequency)
- 3. Avalanche size

We would like to begin by addressing the terminology. We agree that frequency distribution is more appropriate in this context than frequency alone. Accordingly, we will update the manuscript to use frequency distribution of snowpack stability when referring to the overall distribution across all stability classes and frequency when discussing a single class, such as the frequency of stability class poor. While frequency of snowpack stability is the current official term

within the EAWS, we will clarify in the text that we have updated this terminology for precision and context.

We acknowledge that determining the exact frequency distribution of snowpack stability for a region — especially for areas larger than a single mountain slope — is practically impossible. However, it remains the ideal goal, and regional avalanche forecasters strive to assess it as realistically as possible. While modeling may not completely resolve this challenge, it offers the potential to provide more realistic spatial-temporal estimates than individual field observations alone. We believe that the professional framework for determining regional avalanche danger levels should reflect what theoretically needs to be assessed rather than obscuring these requirements. For this reason, we will retain the technical and statistical definition of frequency distribution and clarify that the sum of the frequencies for the four stability classes equals 1. Additionally, we will emphasize the gap between what we aim to measure and what is practically achievable, highlighting the irreducible uncertainties inherent in avalanche forecasting. In this context, clear guidelines on managing incomplete information and addressing resulting uncertainties, can help forecasters to tackle these challenges.

Lines 27-29: describe the preparation of regional avalanche forecasts as involving two steps: 1) assessment of current and future avalanche danger, and 2) communication of future avalanche danger. This is true for the assessment of avalanche danger, but not for the preparation of a regional avalanche forecast, which also includes (depending on region) analysis of snowpack, terrain, locations and avalanche problems. Avalanche danger is only one component of a regional avalanche forecast and this paper focusses on the assessment of avalanche danger ratings. Please revise this section to reflect a broader view of what preparing a regional avalanche forecast entails or narrow the scope of the preparation down to avalanche danger ratings only. This same issue is found on Line 67, where "regional avalanche forecasting" should be narrowed in scope to "the assessment of avalanche danger ratings" and Line 72 where "the process of regional avalanche forecasting" should also be narrowed in scope to "the standards for assessing avalanche danger ratings".

We re-phrased this section and narrowed the scope to assessing regional avalanche danger ratings.

**Lines 47-49:** describe the importance of reliable avalanche forecasts in reducing damage and loss, enhancing safety and mitigating risks. This is all correct but focuses solely on negative consequences and ignores positive outcomes, which is half of the risk equation and a major reason for public avalanche forecasts. It is not all about loss. In addition to preventing loss, avalanche forecasts enable backcountry experiences by highlighting good conditions and appropriate terrain choices, improving the experience for backcountry users. It is essential to not only on focus on mitigation and loss, but also on gain and enabling the backcountry experience. I would like to see this theme applied throughout this paper where appropriate.

We will add a reference to the positive effects of avalanche warnings.

**Line 177:** a better heading would be "Factors and workflow to determine avalanche danger levels"

Changed as suggested to "Factors and workflow to determine avalanche danger levels".

**Tables 2,3 & 4:** I realize these are EAWS standards and perhaps not easily changed, but I suggest you consider the following to make these crucial reference tables more applicable in practice.

These tables should provide enough reference to make assessments based on observational evidence.

**Table 2** - In general I find this table insufficient for making a proper assessment of snow stability. EAWS 2022b Appendix A has much more detail and I suggest improving this paper's Table 2 by adding additional columns to aid in assessment (i.e., stability test results). – Make it clear that this is a point scale assessment. – Change the Description of Very Poor to "very easy to trigger (e.g., natural)" to be consistent

We added the reference to the full table in the caption, but only reprint the short version here for brevity. The updated caption is "Snowpack stability classes refering to the point scale, and the type of triggering typically associated with these classes. For the full table, including typical observations related to each class see EAWS (2022b, Table 1)". We changed the description as suggested to "very easy to trigger (e.g., natural)".

**Table 3** – Consider showing the frequency equation f (i) = ni /n and add a column on the right showing typical ni /n values.

Within the working group, we discussed in depth whether typical values should be shown together with these definitions. In the end, we decided including some examples in the EAWS-document describing the Matrix (Appendix B in EAWS, 2022b). However, the results shown there are based on one or two services and from very confined regions. In addition the values diverge substantially between stability tests and avalanche activity. We therefore concluded that they may be more confusing than clarifying. Therefore, we prefer not to show exemplary frequency values in this table.

**Table 4** – Rephrase Size 1 be more concise, "Unlikely to bury or injure a person except in terrain traps".

This is taken from a standard. We only recite it here and will not change it. But we will take this comment to the appropriate forum.

**Line 430** – The SWI forecasters were instructed to assess the three contributing factors, but to then ignore the EAWS Matrix and this led to a 46% deviation from the EAWS Matrix. This is a substantial deviation and makes me wonder why. Please explain why there is such a large deviation. Also Line 433 says that "forecasters largely concurred with the danger level proposed by the EAWS Matrix", except for SWI, which is the only service not linked to the matrix, where they disagreed 46% of the time. I think this needs more explanation please.

In most of the analyzed warning services, forecasters utilized the EAWS Matrix to determine the danger level (DL), typically aligning their choice with the first DL suggested by the Matrix. Deviations from this initial DL were relatively rare in these services (see also Table 6). In Switzerland, however, forecasters assessed factors and DL independently, without directly referencing the EAWS Matrix to determine the DL. This approach is somewhat like the North American system, where factors (e.g., *likelihood* and *avalanche size*) are evaluated for each avalanche problem, but where the final DL decision remains with the forecaster as no clear assessment rules are defined (Clark and Haegeli, 2018). While Swiss forecasters frequently deviated from the first DL shown in the EAWS Matrix, absolute differences were generally minor as forecasters most frequently selected the sub-level closest to this DL (for description of sub-levels used in Switzerland refer to Techel et al. 2022). These findings are described in greater

detail, analyzing the second season after introducing the Matrix, in Techel et al. (2024). We will provide these explanations in the Discussion section.

**Line 443:** This is just a comment, but the fact that "All warning services described stability most often as poor when giving "D=2" is interesting because typically at D2, avalanche conditions are generally stable except for a few locations, thus backcountry travel can be done in many locations. This scenario shows that forecasters are assessing this D2 based on the frequency of poor stability. Although this Matrix applies the concept of snow stability, the assessments of avalanche danger ratings are actually determined and driven by the frequency of instability. This links to my earlier comments about Lines 47-49 whereby this model is driven by negative consequences, which is a bias towards instability and fails to appreciate stable conditions. No change requested, just an observation which has followed me through this review and has made me wonder why stability is not instead referred to as instability, when this is what truly drives the D assessment?

This issue has been discussed within the Matrix workgroup and among EAWS members. The term stability works better in communication and direct speech, which to our knowledge also true for the North American avalanche community. We would like to stick to the term stability. However, we generally agree with your observation and concerns. We hope and agree that an avalanche forecast has positive effects as you mention. However, the main task of an avalanche forecaster is to identify the danger and where/when it is located.

**Technical corrections:**

Line 6: Changed to "promote" as suggested

**Line 19:** Changed to "...to improve the reliability, credibility and timeliness of avalanche forecasts,..."

**Line 44:** changed to "forecaster's" **Line 88:** changed to "Except" **Line 108:** changed to "frequency" **Line 131:** replaced reference a suggested

Line 158: fixed

**Line 220:** fixed **Line 298:** We added information on terrain: "The workflow is specifically designed for regional avalanche forecasters. It assumes that the forecast area is large enough to encompass multiple mountains, elevation zones, all aspects, and varied terrain features, such as ridges, gullies, and open slopes. Consequently, terrain is not treated as an independent factor."

Figure 3 caption: We replaced "the most vulnerable locations" by "the lowest stability class".

**Line 306:** Rephrased to "...we aggregated bordering warning services to groups of services when using the same workflow and forecasting software."

Line 561: We replaced "imminent" by "crucial". Line 635: Changed as suggested.

**References**

Clark, T. and Haegeli, P.; ESTABLISHING THE LINK BETWEEN THE CONCEPTUAL MODEL OF AVALANCHE HAZARD AND THE NORTH AMERICAN PUBLIC AVALANCHE DANGER SCALE: INITIAL

EXPLORATIONS FROM CANADA; Proceedings, International Snow Science Workshop, Innsbruck, Austria, 2018

EAWS: Determination of the avalanche danger level in regional avalanche forecasting, European Avalanche Warning Services, <a href="https://www.690">https://www.690</a> avalanches.org/wp-content/uploads/2022/12/EAWS\_matrix\_definitions\_EN.pdf, last access: 21 June 2023, 2022b.

Techel, F., Mayer, S., Pérez-Guillén, C., Schmudlach, G., and Winkler, K.: On the correlation between a sub-level qualifier refining the danger level with observations and models relating to the contributing factors of avalanche danger, Natural Hazards and Earth System Sciences, pp. 1911–1930, https://doi.org/10.5194/nhess-22-1911-2022, 2022

Techel, F., Lucas, C., Müller, K., Pielmeier, C., and Morreau, M.; UNRELIABILITY IN EXPERT ESTIMATES OF FACTORS DETERMINING AVALANCHE DANGER AND IMPACT ON DANGER LEVEL ESTIMATION USING THE MATRIX; Proceedings, International Snow Science Workshop, Tromsø, Norway, 2024

Schweizer, J., Kronholm, K., Wiesinger, T.; Verification of regional snowpack stability and avalanche danger; Cold Regions Science and Technology, Volume 37, Issue 3, Pages 277-288, <a href="https://doi.org/10.1016/S0165-232X(03)00070-3">https://doi.org/10.1016/S0165-232X(03)00070-3</a>; 2003.

Bakermans L, Jamieson B, Schweizer J, Haegeli P. Using stability tests and regional avalanche danger to estimate the local avalanche danger. Annals of Glaciology. 2010;51(54):176-186. doi:10.3189/172756410791386616

**Reply to Erich Peitzsch**

We sincerely thank Erich Peitzsch for his thorough and constructive review. We tried to address each point raised below.

**General comments**

In this study, the authors provided a thorough background on the European Avalanche Danger Scale (EADS) and European Avalanche Warning Services (EAWS) Matrix development and definitions, described the revised EAWS Matrix and associated methods of updating the Matrix, and then presented relevant results of using the newly revised EAWS Matrix for one full season. This manuscript is well written and logically organized. The methods are sound, and the results are supported by sufficient evidence. The interpretation of those results is reasonable with reference to existing literature. The recommendations section of this paper is very useful and points to limitations of consistency by providing solutions. Overall, I think this is a valuable contribution to the literature, fit well within this Special Issue, and should be published. I have a couple of general comments for the authors to consider and a few specific and technical comments as well.

I understand that this study focuses on evaluating the updated (current) EAWS matrix, its use and consistency among forecasters, and compatibility with EADS, and not necessarily an evaluation of the individual three key factors that determine the danger rating. As the authors point out, consistency in forecasters' evaluation or interpretation of the three key factors is crucial for the

matrix to be used to its full potential. Indeed, the authors provide recommendations on how to enhance the use of the matrix by improving consistency in the three contributing factors. However, it seems that across the surveyed regions, a variety of input data (in-situ observations, model output, meteorological data, etc.) exist. The authors mention the influence of input data very briefly (lines 485-489), but the quality and quantity of input data plays a crucial role in danger assessment and would potentially influence the assessment variability across forecasters and regions. Can the authors provide information or comment on, generally, what data types each region uses and/or provide evidence on how the assimilation process of various types of data across regions may influence the classification of the three factors and ultimately the danger rating?

We agree that the type and availability of relevant data and their spatial and temporal density likely influences the assessment. To our knowledge, field observations, measurements from automatic weather stations, and weather- and snowpack models are used to varying degrees in different warning services. As we have not collected data regarding this, we are not able to relate availability of various data sources to Matrix usage. While this could be interesting to investigate in a future study, here we will not speculate on this.

The authors present a thorough summary on the evolution of EAWS and the EADS and provide some geographic references to other non-European forecasting tools like the CMAH in North America. However, there are no references to other avalanche sector decision making tools. In other words, this study focuses on public avalanche forecasting operations, but not forecasting in other sectors like transportation corridors, ski areas, natural resource industry, etc. Is the same tool used for those sectors throughout Europe? I suggest being clear that this study focuses on a matrix for public avalanche forecasting or state how using the EAWS matrix in those sectors differs, if at all, from public avalanche forecasting.

You are right. The EAWS Matrix is solely applied in regional avalanche forecasting and not in other operations such as guiding. We will update the introduction and background accordingly and emphasize that we focus on regional avalanche forecasting.

**Specific and Technical Comments**

**Figure 2:** Consider adding a legend to the proportion scales for b) and c) that easily shows the reader which colors represent higher correlation values.

On purpose, we avoided adding a legend to keep the figure compact but added actual values in the cells instead. However, we noted that the explanation in the caption "Stronger color saturation indicates a larger proportion of responses." is erroneous. We will update the figure caption to "Cells with stronger color saturation indicate cells with lower agreement (b) or fewer responses (c)."

**Figure 6:** The use of D1 and D2 here is confusing. Is this the same as the median D1 and D2 used in Figure 2a and defined in Lines 262 -264? Also, if there was disagreement with D (forecaster derived) and D1 (Matrix derived), that is indicated in the left column in Figure 6, correct? If the second column represents D1  $\neq$  D2, then forecasters used D2 (again the median second selection from Figure 2a)? Please clarify.

Yes, D1 refers to the majority voting for a D and D2 to a second D in case of more than 25% of the votes (interquartile range) in both cases. We can update the caption to clarify that. E.g.,

"Proportion of cases that D=D1 or D=D2 was used by groups of warning services (center column)."

**Line 88:** four? In "Despite for minor changes in 1994..." or do you mean "Except for minor changes..."

We will change to "Except"

**Line 104:** the way this is written is confusing to me. I read it as 'as stability decreases you need a greater load to trigger an avalanche.' (i.e. inverse relationship). Perhaps 'instability' should be used here instead of stability.

The point we wanted to make here was that both columns, on snowpack stability and likelihood of triggering, describe the same phenomena and are to some extent redundant. We will remove the word "inversely" and state that "The load necessary to trigger an avalanche is correlated to snowpack stability".

**Line 241 & 244:** 60 responses in total in line 241, but in line 244, you state 76 responses. What is the difference?

60 was the number of responses in our survey. In addition, we considered the responses from the EAWS working group members in 2019 and in 2022, which added 14 votes. The remaining two votes came from the referenced literature making a total of 76. So 60 votes from an online survey among EAWS forecasters and 16 votes from outside the online survey. We list the sources in the bullet points in line 236-243.

**Line 258:** Similar to the comment above (Line 104). I view an increase in stability as the snowpack becoming more stable. Instability?

We will rephrase "Examining the most frequently selected combinations confirms that stability, frequency, and size tend to increase with higher danger levels." to "The most frequently selected combinations confirm that snowpack stability decreases, while the frequency of the respective stability class and avalanche size tend to increase with increasing avalanche danger level."

**Line 345:** "center of gravity"? Do you mean largest proportion?

We will rephrase "D = 2 (n = 6806, 56%) has its center of gravity at poor stability with a few or some locations and avalanche size 2." to "D = 2 (n = 6806, 56%) is primarily clustered around poor stability, with a few or some locations and avalanche size 2."

---

## Author Response (AR2)

Dear Pascal,

Thank you very much for your thorough and constructive feedback on our manuscript. We fully agree with your suggestion to split the paper into two parts, and we are confident that this will improve readability and allow us to better align with the expected structure of a scientific publication.

We have now split the original manuscript into two separate papers with the following titles:

- The EAWS Matrix, a Look-up Table to Determine the Regional Avalanche Danger Level (**Part A): Conceptual Development**
- The EAWS Matrix, a Look-up Table to Determine the Regional Avalanche Danger Level (Part B): Operational Testing

The current resubmission represents **Part A**, which focuses on the conceptual development of the EAWS Matrix and the analysis of the expert survey. It also includes references to key findings from the operational analysis (Part B) and related consistency studies (presented and published at ISSW 2024) to provide context and support the discussion. **Part B** will be submitted separately by Frank. Unfortunately, we were unable to finalize Part B this week, but we aim to submit it early next week. We hope it will be possible for you to handle both parts together once they are available.

Given the substantial restructuring of the manuscript (see latexdiff), we hope it is acceptable that we do not provide a detailed point-by-point response to your original editorial comments. Instead, we summarize below the key changes we have made in response to your suggestions:

- We provide a more structured overview of existing standards (e.g., the danger scale, conceptual model and matrix tools).
- We presented the definitions of the Matrix factors and provided a detailed description of the expert survey and its analysis.
- The revised manuscript references the two related studies: one on consistency presented at ISSW 2024 and one submitted as a companion paper to NHESS, which focuses on operational usage. Key findings from both are summarized in the discussion and conclusion of Part A.
- We revised and extended the sections discussing the relationship between the CMAH and the EAWS Matrix, aiming for greater clarity and conceptual alignment.
- We removed the section on the updated danger scale and the more speculative recommendations to keep the manuscript focused. The development of the revised danger scale is ongoing and may be presented in a separate publication at a later stage.

The data supporting both companion *NHESS* manuscripts will be made available on EnviDat.org once Part B has been submitted.

We hope the revised manuscript meets your expectations and addresses your feedback appropriately. Thank you again for your support and guidance throughout this process.

Best regards, Karsten, Frank, and Christoph

---

## Author Response (AR3)

Dear Pascal,

Thank you very much for your effort and constructive feedback on our manuscript. We believe your input helped us improve the paper substantially.

We have implemented most of your comments, and the revised version reflects both structural and content-related improvements.

**Regarding your major comments:**

We have slightly restructured the manuscript. It now consists of:

- an updated Introduction (Sec. 1) and Background (Sec. 2),
- a clear documentation of the three development steps leading to the current version of the Matrix (Sec. 3 to 5),
- a new section explaining how the Matrix is intended to be used, including a practical example, a discussion of existing limitations and the relationship to the CMAH (Sec. 6),
- and a conclusion (Sec. 7).

We hope this structure makes the paper more accessible to practitioners—allowing them to go directly to section 6—while still thoroughly documenting the development process.

We did not modify the definitions, as they have been adopted as EAWS standards. To avoid confusion, we chose to keep them as published. However, we have added further explanations and clarifications where possible. The tables containing typical observations and the remarks from the EAWS documentation are now included in the appendix to serve as a complete reference within the paper.

**Regarding your technical comments:**

We have addressed all of them as suggested. Please refer to the latexdiff for a detailed overview of the changes.

Thank you again for your valuable feedback and support throughout this process.

Best regards,

Karsten, Frank, and Christoph